# A Measure of the Complexity of Neural Representations based on Partial Information Decomposition

**David A. Ehrlich**[1,2,*]                                         *davidalexander.ehrlich@uni-goettingen.de*
**Andreas C. Schneider**[3,2,*]                                         *andreas.schneider@ds.mpg.de*
**Viola Priesemann**[2,3,4]
**Michael Wibral**[1,4]
**Abdullah Makkeh**[1]

[1] *Goettingen Campus Institute for Dynamics of Biological Networks, University of Goettingen, 37077 Goettingen, Germany*
[2] *Max Planck Institute for Dynamics and Self-Organization, 37077 Goettingen, Germany*
[3] *Institute for the Dynamics of Complex Systems, University of Goettingen, 37077 Goettingen, Germany*
[4] *Campus Institute Data Science, University of Goettingen, 37077 Goettingen, Germany*

**Reviewed on OpenReview:** *https://openreview.net/forum?id=R8TU3pfzFr*

## Abstract

In neural networks, task-relevant information is represented jointly by groups of neurons. However, the specific way in which this mutual information about the classification label is distributed among the individual neurons is not well understood: While parts of it may only be obtainable from specific single neurons, other parts are carried redundantly or synergistically by multiple neurons. We show how Partial Information Decomposition (PID), a recent extension of information theory, can disentangle these different contributions. From this, we introduce the measure of "Representational Complexity", which quantifies the difficulty of accessing information spread across multiple neurons. We show how this complexity is directly computable for smaller layers. For larger layers, we propose subsampling and coarse-graining procedures and prove corresponding bounds on the latter. Empirically, for quantized deep neural networks solving the MNIST and CIFAR10 tasks, we observe that representational complexity decreases both through successive hidden layers and over training, and compare the results to related measures. Overall, we propose representational complexity as a principled and interpretable summary statistic for analyzing the structure and evolution of neural representations and complex systems in general.

---

*These authors contributed equally to this work.

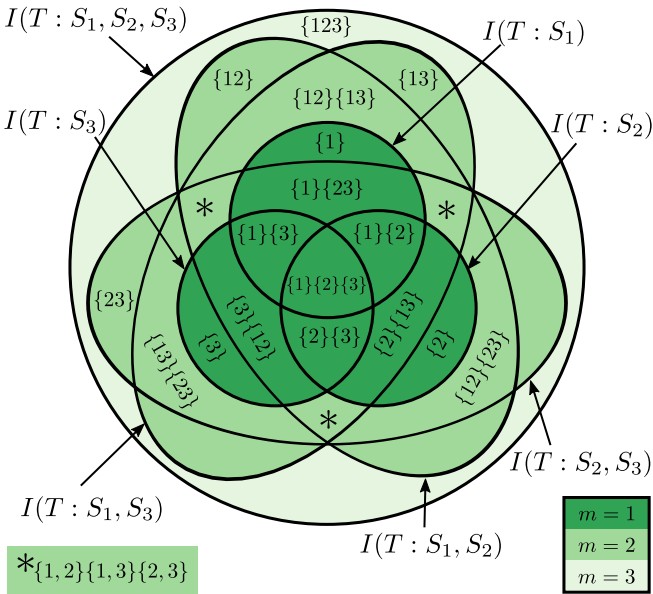

Figure 1: **Mereological diagram illustrating how the PID atoms $\Pi(T : \boldsymbol{S}_\alpha)$ constitute all classical mutual information quantities involving three source variables, as introduced by Williams & Beer (2010).** Classical mutual information terms are represented by black ovals with general redundancies given by their overlaps. The PID atoms correspond to the contiguous areas of the overlaps and are labeled by their corresponding antichains $\alpha$. The green shading indicates the degree of synergy $m$ of the respective atoms.

# 1 Introduction

Despite their tremendous success, the inner workings of artificial neural networks remain mostly elusive to this date (LeCun et al., 2015; Alom et al., 2019; Samek et al., 2017). It is known that, since all of the information a network can use to solve a task must already be contained in the model's inputs, the purpose of hidden internal components remains to distill the pertinent features and make them available to later computational steps. However, the *way* in which the features are represented among multiple neurons and how this representation changes throughout the training phase is not well understood (Li et al., 2015; Bengio et al., 2013). This lack of understanding continues to pose a major obstacle to the adoption of machine learning techniques in critical settings, such as medical image analysis or self-driving cars, but also hinders principled development of better machine learning algorithms (Samek et al., 2017; Amodei et al., 2016; Eykholt et al., 2018).

In a classification problem, the difficulty of extracting information about the target label from internal representations in hidden layers is a key factor for network performance and interpretability alike. As a first step towards these objectives, one needs to define and quantify this notion of *difficulty*, which originates from complex higher-order statistical interrelations between neurons. These relations – if present – make it necessary to consider large groups of neurons at a time to be able to discriminate between certain represented target values. The number of neurons minimally needed to obtain a piece of target-discriminatory information is conceptually related to the classical question of how many neurons are active in the representation of an input sample, and may thus be viewed as an information-theoretic equivalent of a sparsity measure. We propose to quantify this 'information sparsity' in a principled way by answering an intuitive question: On average, how many neurons do I need to observe simultaneously in order to gain access to a piece of information? Note that while we limit ourselves to classification problems in this paper, these questions pertain to a vast class of problems not limited to machine learning systems wherever information about some target variable is distributed amongst a set of equivalent information bearing units.

Note that while multiple notions of information exist in literature (e.g. semantic information that has meaning for a biological or technical system) in this paper the term *information* is used in a narrow sense referring to Shannon information only. Originally developed for the analysis of communication channels, Shannon's Information Theory (Shannon, 1948) is a framework capturing general, possibly non-linear correlations between random variables. As such, it lends itself to the analysis of neural network as a high mutual information between the classification label and latent representations can be proved to be a necessary requirement for good classification accuracy via Fano's inequality in conjunction with the data processing inequality (Geiger, 2021, Appendix IV). The choice of Shannon Information over other non-semantic measures such as Renyi Entropy (Rényi, 1961) or Tsallis Entropy (Tsallis, 1988) has been made since the cross entropy loss function that is ubiquitous in classification networks uses Shannon information terms (Goodfellow et al., 2016), making it plausible that tracking information theoretic quantities can provide an insight into the learning dynamics.

From the perspective of information theory (Shannon, 1948), neural networks are just information channels that forward the task-relevant part of the input to the output layer while shedding task-irrelevant parts (Tishby & Zaslavsky, 2015). This said, the idea of using Shannon information to quantify the information in the activations of Deep Neural Network (DNN) hidden layers has been met with fierce debate after methodological flaws surfaced (Saxe et al., 2019; Goldfeld et al., 2019) in the influential early works by Tishby et al. (Tishby & Zaslavsky, 2015; Shwartz-Ziv & Tishby, 2017). We address these issues in this work and show how information theory can be applied to quantized networks in a theoretically sound way (Section 2).

Yet, even with such methodological problems out of the way, the question posed above on how features are represented cannot be answered using Shannon mutual information alone. This is due to the inability of pairwise information measures to disentangle redundant and synergistic contributions between multiple neurons (Williams & Beer, 2010), leading to inevitable double- or under-counting of information contributions. Non-overlapping information contributions can be obtained by decomposing the mutual information $I(T : \boldsymbol{S})$ between a target random variable $T$ and several source variables $\boldsymbol{S} = (S_1, \ldots, S_n)$ into its information *atoms* $\Pi(T : \boldsymbol{S}_\alpha)$ such that

$$I(T : \boldsymbol{S}) = \sum_\alpha \Pi(T : \boldsymbol{S}_\alpha), \tag{1}$$

which has been made possible by the introduction of a recent extension to information theory known as Partial Information Decomposition (PID) (Williams & Beer, 2010). These atoms are the smallest set of quantities from which the mutual information between $T$ and all possible subsets of $\boldsymbol{S}$ can be combined (Figure 1) (Gutknecht et al., 2021). Note that here, each information atom is referred to by its corresponding 'antichain' $\alpha$, which will be explained in Section 3.1.

From the PID atoms, we can answer the question of how many neurons are needed on average to access a piece of information in a principled way by introducing the summary statistic we term "Representational Complexity". A representational complexity of $C = 1$ means that all information can be obtained from single neurons, while a representational complexity $C$ that is equal to the number of neurons in a layer means that no information can be obtained about the target unless one has access to all neurons. By tracking the representational complexity $C$ of the hidden layers over training we shed light upon a key aspect of how the internal representations of the target variable evolve during the training phase. This understanding may subsequently be employed to improve network designs by providing a tool to compare the dynamics in different network architectures.

The main contributions of this work are **(i)** describing a principled approach for applying PID to analyze the representations in DNNs and related systems, **(ii)** the introduction of representational complexity as a measure of information sparsity, **(iii)** discussing subsampling and coarse-graining procedures for the estimation of representational complexity and proofs of bounds on the latter, **(iv)** empirical results in quantized DNNs showing a decrease in representational complexity over training and through successive hidden layers, **(v)** comparisons of the empirical results to two related approaches and **(vi)** empirical results showing a difference between representational complexity computed on the train or test dataset.

## 2 Related works

Tishby & Zaslavsky (2015) were among the first to attempt to analyze deep neural networks from an information-theoretic perspective. Information theory, which was developed for the analysis of noisy information channels (Shannon, 1948), allows to quantify information in an observer-independent way and helps to form a clear criterion of relevancy of information in hidden representations by distinguishing between the mutual information of layer activations with the ground truth classification labels and with the input variables. Viewing the networks as a sequence of such channels, Shwartz-Ziv & Tishby (2017) computed mutual information from *binned* activations of the individual hidden layers, and plotted the resulting trajectory in what they termed the *information plane*. This information plane is a two-dimensional space with the mutual information between hidden layer activations and input, and between hidden layer activations and ground-truth label as its two dimensions.

However, their claim to estimate actual mutual information quantities of the network variables with their binning approach was later shown to be unfounded (Saxe et al., 2019; Goldfeld et al., 2019). In fact, the true mutual information between the network's inputs or label and a hidden layer is either infinite, in the case of continuous features, or constant and equal to the finite entropy of the discrete inputs or labels (Saxe et al., 2019; Goldfeld et al., 2019; Geiger, 2021). This is due to the fact that the network itself defines a deterministic and almost always injective mapping from inputs to hidden layer activation values. For this reason, the results shown by Shwartz-Ziv & Tishby (2017) do not constitute estimates of actual information-theoretic quantities and may at best be reinterpreted as measures for geometric clustering (Goldfeld et al., 2019; Geiger, 2021).

Building on this realization, Goldfeld et al. (2019) showed that a meaningful information-theoretic analysis can be performed by disrupting the injectivity and limiting the channel capacity of the network forward function. While they achieve this by explicitly adding Gaussian noise to each activation value, we achieve the same goal by training and analyzing quantized very-low precision activation values in this work. This approach is in line with recent trends towards low (Gupta et al., 2015) and very-low precision (Hubara et al., 2017) computing for reasons of efficiency and scalability. The crucial difference to Shwartz-Ziv & Tishby (2017)'s binning approach lies in the fact that the quantization we employ is intrinsic to the network itself, making mutual information quantities well-defined and meaningful as well as ensuring the data processing inequality holds for the Markov chain of successive hidden layers.

The picture of artificial neural networks as mere information channels needs to be refined following the insight that network layers need not only to pass on all relevant information but also to transform it in such a way that it becomes accessible for subsequent processing. To understand this representation of information within each layer, one needs to go beyond analyzing hidden layers as a whole and look at the structure of information representation across the neurons within those layers instead, which can be done in a principled way by employing partial information decomposition.

Previous works on PID in artificial neural networks have used PID to motivate and interpret classical mutual information quantities (Wibral et al., 2017) and used it to analyze filters in convolutional neural networks (Yu et al., 2020). Furthermore, Tax et al. (2017) used PID to analyze pairs of neurons in generative neural networks and find that the networks move towards more unique representations of the target in later stages of training. We expand upon this approach by being the first to analyze all neurons of a layer as individual PID sources, which allows also to uncover higher-order interactions between them.

Closely connected to our approach, Reing et al. (2021) derived scalable measures for quantifying higher-order interactions between neurons using Shannon information. By focusing more on scalability, however, the authors trade in some of the interpretability and expressivity of an approach based on PID. Furthermore, Reing et al. (2021) cover mostly the analysis of representations without reference to a target variable, while we focus on analyzing only task-relevant information contributions. An empirical comparison to this approach can be found in Section 4.3.

In the context of representation learning, information-theoretic approaches focus mostly on quantifying the degree of entanglement of latent dimensions in Variational Autoencoders, often with respect to the information about some underlying generative factor. This is done by measuring the total correlation (Kim & Mnih, 2018) as a summary statistic or the difference between two classical mutual information quantities (Chen et al.,

2018; Tokui & Sato, 2021). These works are related to ours, but analyze only a specific layer, do not use the label information as target and stay within the realm of classical Shannon information framework.

Other works on representations in deep neural networks have attempted to define notions of *usable* information based on the idea of restricting the ability of an observer to perfectly decode the presented information (Xu et al., 2019; Kleinman et al., 2020). A related approach is the probing of representations using simple linear readouts, for example Alain & Bengio (2017), who find empirical evidence hinting at less complex representations in deeper layers. Additionally, several other summary statistics of representations have been suggested based on dimensionality analysis (Ansuini et al., 2019; Fukunaga & Olsen, 1971) or canonical correlation analysis (Morcos et al., 2018; Kornblith et al., 2019). We compare an analysis based on local intrinsic dimension to our measure in Section 4.4.

Several other complexity measures were compared by Jiang et al. (2019), who analyzed them in terms of their ability to predict the networks' capacities for out-of-sample generalization. This idea is founded on the intuitive notion that less complex representations should be more robust to minor changes in the inputs. As our paper is focused on interpretability of the novel measure of representational complexity, potential ties to generalization ability have not been studied as of yet.

## 3 Deriving an interpretable measure for the complexity of a representation from partial information decomposition

### 3.1 Background: Partial information decomposition

The mutual information $I(T : \boldsymbol{S}) = I(T : S_1, \ldots, S_n)$ that several source random variables $\boldsymbol{S} = \{S_1, \ldots, S_n\}$ carry about a target $T$ can be distributed amongst these sources in very different ways. While some pieces of information are *unique* to certain sources, others are encoded *redundantly* by different sources, while yet others are only accessible *synergistically* from several sources considered jointly. With three or more sources, even more complex contributions, in general describing redundancies between synergies, emerge. For example, some information about $T$ might be accessible either from source $S_3$ alone, or – redundantly to this – from a synergistic combination of sources $S_1$ and $S_2$, but from nowhere else; this information would constitute one of the information atoms $\Pi$.

As mentioned before, these information atoms can be combined to form all classical mutual information quantities $I(T : \boldsymbol{S_a})$ between $T$ and subsets of sources $\boldsymbol{S_a} = \{S_i | i \in \boldsymbol{a}\}$ with indices $\boldsymbol{a} \subseteq \{1, \ldots, n\}$. Conversely, the information atoms can be uniquely identified by which classical mutual information quantities they contribute to. Mathematically, this notion is captured by their corresponding *parthood distribution* $\Phi : \mathcal{P}(\{1, \ldots, n\}) \to \{0, 1\}$ – a binary function defined on the powerset $\mathcal{P}$ of source indices that is equal to "1" for exactly those sets $\boldsymbol{a}$ of source indices for which the atom $\Pi(T : \boldsymbol{S_\Phi})$ is part of the mutual information $I(T : \boldsymbol{S_a})$ (Gutknecht et al., 2021, Section 2) (for a detailed explanation, see Appendix A.1.1). Thus, the mutual information of any set of sources $\boldsymbol{S_a}$ and $T$ can be written as

$$I(T : \boldsymbol{S_a}) = \sum_{\{\Phi | \Phi(\boldsymbol{a}) = 1\}} \Pi(T : \boldsymbol{S_\Phi}). \tag{2}$$

Instead of labelling the atoms by their parthood distribution $\Phi$, the atoms can equivalently be referenced as $\Pi(T : \boldsymbol{S_\alpha})$ using certain sets of sets of source indices $\alpha \in \mathcal{P}(\mathcal{P}(\{1, \ldots, n\}))$, which can be mapped one-to-one to parthood distributions (Gutknecht et al., 2021) and are referred to as *antichains* by Williams & Beer (2010), who first introduced PID using these antichains (for a detailed explanation, see Appendix A.1.2). The antichains make apparent the connection between atoms and their meaning as redundancies between synergies: For example, the atom from before capturing the information that can be obtained either from $S_3$ alone or synergistically from $S_1$ and $S_2$ together is referred to by the antichain $\{\{1, 2\}, \{3\}\}$.

The number of atoms scales super-exponentially with the number of sources $n$, increasing from 7579 for $n = 5$ sources to over 7.8 million for $n = 6$ (Williams & Beer, 2010; Wiedemann, 1991). Note, however, that this increase in the number of atoms should not be mistaken for a shortcoming of PID but rather as

an acknowledgment of the vast number of configurations information can be encoded in among multiple variables.

On the other hand, there are only $2^n - 1$ mutual information quantities that provide constraints through Equation (2). One way to resolve this underdeterminedness is through the introduction of a measure for redundancy $I_\cap(T : \mathbf{S}_\alpha) = I_\cap(T : \{\mathbf{S}_{\mathbf{a}_1}, \ldots, \mathbf{S}_{\mathbf{a}_k}\})$ between collections of sources indexed by $\mathbf{a}_i \subseteq \{1, \ldots, n\}$. Noting that mutual information can be interpreted as a "self-redundancy" such that $I(T : \mathbf{S}_{\mathbf{a}}) = I_\cap(T : \{\mathbf{S}_{\mathbf{a}}\})$, these redundancies can be constructed from the information atoms in an analogous and consistent way to Equation (1) as

$$I_\cap(T : \mathbf{S}_\alpha) = \sum_{\beta \preceq \alpha} \Pi(T : \mathbf{S}_\beta), \tag{3}$$

where $\preceq$ refers to the partial order of antichains on the redundancy lattice (for a detailed explanation, see Appendix A.1.3) (Crampton & Loizou, 2000; 2001; Williams & Beer, 2010). Since now the number of defining equations is equal to the number of atoms, these can be computed by inverting Equation (3), which is known as a *Moebius-Inversion* (Rota, 1964; Williams & Beer, 2010).

Over recent years, a number of different redundancy measures have been suggested which fulfill a multitude of different additional desiderata (Lizier et al., 2018), e.g., from decision theory (Bertschinger et al., 2014), game theory (Ince, 2017) or Kelly gambling (Finn & Lizier, 2018). In this paper, we utilize the $I_\cap^{\mathrm{sx}}$ measure introduced by Makkeh et al. (2021), where 'sx' stands for shared exclusions of probability mass. In essence, the measure defines redundancy by the regions of probability space which are jointly excluded by observing the realizations of multiple collections of random variables. As this redundancy measure draws only on notions from probability theory it is the most canonical choice for our purpose of analyzing neural networks. The concept of representational complexity that we introduce here, however, can readily be generalized to any other redundancy-based multivariate PID.

### 3.2 Representational complexity

In order to gain insight into the representation of task-relevant information in multiple equivalent source variables, we introduce in the following a principled way to evaluate the question: On average, how much of a system needs to be observed simultaneously to access a particular piece of information? We propose that this difficulty of retrieving label information may be quantified by considering the minimum number of sources that are needed jointly in order to reveal a piece of information. As a first step towards this goal, one needs to dissect the total mutual information into pieces which individually have a clear notion of how many neurons are needed to retrieve the information.

Note, however, that such a dissection cannot be achieved with mutual information due to its inability to disentangle synergistic and redundant contributions: As an example, take two source variables $S_1$ and $S_2$ which carry information about a target $T$. Since the mutual information term $I(T : S_i)$ captures all information that the single source $S_i$ carries about $T$, one might think that the total information that can be obtained from single sources is given by the sum $I(T : S_1) + I(T : S_2)$. However, apart from their individual unique contributions, $S_1$ and $S_2$ might carry some identical pieces of information about $T$, and this redundancy is double-counted in the above sum. Moreover, the two sources might carry some information about $T$ in a way that it is only accessible from both sources taken together, e.g., if one source represents a cipher text with the second providing the decryption key. The joint mutual information $I(T : S_1, S_2)$ thus consists of four atoms: The two unique atoms of the respective sources, their redundancy and their synergy, while the individual mutual information terms $I(T : S_i)$ consist of the unique information of the respective source and the redundancy between both sources. Thus, if we subtract the sum of the individual mutual information terms from the joint term we obtain

$$I(T : S_1, S_2) - I(T : S_1) - I(T : S_2) = \text{synergy} - \text{redundancy},$$

with no way to quantify synergy or redundancy individually. Since the redundancy can be obtained from any single variable while the synergy necessarily requires access to both variables jointly, it is impossible to

dissect the joint mutual information into pieces with a well-defined minimal number of neurons needed to retrieve the information within the framework of classical Shannon information alone.

The PID atoms, on the other hand, do have a well-defined minimal number of neurons necessary to retrieve the information, which is captured by what we term the "Degree of Synergy" $m$ given by

$$m(\alpha) := \min_{\boldsymbol{a} \in \alpha} |\boldsymbol{a}|, \tag{4}$$

which is illustrated for all trivariate PID atoms in Figure 1. For instance, the PID atom with the antichain $\{\{1\}\{4\}\{2,3\}\}$ has a degree of synergy of $m = 1$ since the information can be obtained by observing the single sources $S_1$ or $S_4$, while the atom with the antichain $\{\{1,2\}\{3,4,5\}\}$ has a degree of synergy of $m = 2$ since the smallest set of sources the information can be retrieved from is the pair consisting of $S_1$ and $S_2$.

Finally, the average degree of synergy, weighted by the relative information contributions of the respective atoms, defines the "Representational Complexity" $C$ as

$$C := \frac{1}{I(T : \boldsymbol{S})} \sum_{\alpha} \Pi(T : \boldsymbol{S}_\alpha) \, m(\alpha). \tag{5}$$

A representational complexity of $C = 1$ means that all information can be obtained from single sources, while $C$ is equal to the number of sources if the information is spread purely synergistically between all of them. Bare in mind that representational complexity is defined as the *average* over PID atoms and does thus not reflect the "worst case": Even if $C$ is smaller than its theoretical maximum one might need more than $C$ sources to identify a specific target label.

An alternate way to view the average in Equation (5) is to first sum up all PID atoms of a fixed degree of synergy $m$ and then taking their weighted average. These sums of PID atoms are conceptually related to the *Backbone Atoms* introduced by Rosas et al. (2020). Rosas measure, however, does not constitute a PID in the strict sense, as mutual information terms between the target and subsets of sources cannot be constructed from the atoms. This makes the backbone atoms less suitable for quantifying how many neurons are needed to access a piece of information.

The concept of representational complexity is linked to the idea of sparse coding in neuroscience (Olshausen & Field, 2004), in that both approaches aim to capture the relevancy of higher-order relations between neurons. While sparsity measures quantify the spread of activity tied to individual realizations across the neuron population, e.g., by measuring the average momentary number of active neurons, representational complexity measures the spread of *information*, i.e., how distributed the ability to *distinguish between* realizations of the target variable (e.g. the label in classification tasks) is. Note further that representational complexity is computed on the mutual information with the network's target variable, thus relating to only task-relevant components of the activation patterns, while sparsity measures are generally not selective about the task-relevancy of the activations.

From its definition as an expectation value of the degree of synergy weighted by the fractional information contribution of the respective PID atoms, it is evident that $C$ is well-defined whenever $I(T : S) \neq 0$, otherwise $C$ is not only mathematically undefined by also conceptually vacuous. Another aspect to pay attention to is that when both the target and the sources are continuous variables, $C$ can be undefined when $I(T : S)$ is infinite. Thus, we can determine that $C$ is well-defined for the quantized activation values with finite alphabets considered in this paper.

From the information-theoretic structure of $C$, it is easy to see that certain desirable properties of mutual information and PID are inherited by $C$.

- **Symmetry** Representational complexity is invariant under the reordering of the source variables.

- **Invariance under isomorphisms** Representational complexity is invariant under isomorphic mapping of the target or the sources, for example, invertible relabelling of classification labels.

There are other desirable properties that are related to the choice of the PID measure. The first is the continuity as a function of the joint probability distribution $p(T, S)$. This is an important prerequisite to a

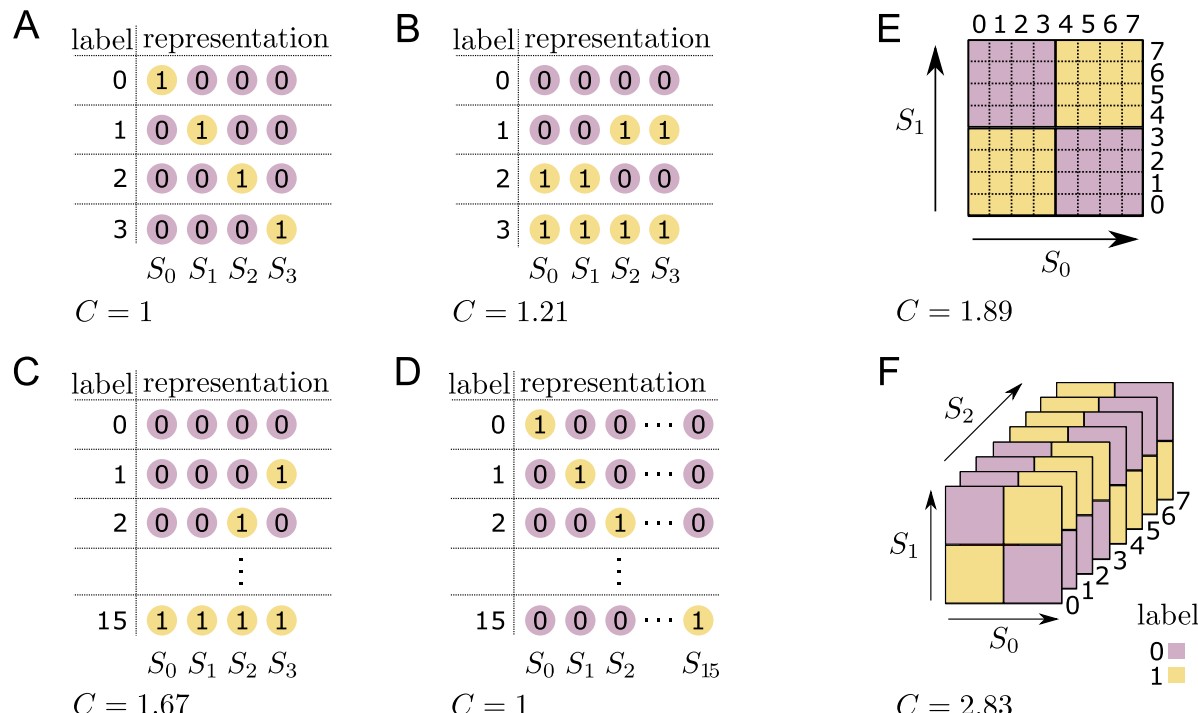

Figure 2: **Demonstration of representational complexity with simple examples.** Subfigures **A-D** show different label encoding schemes in binary neurons $S_1, \ldots, S_n$ (circles). Subfigure **E** shows an encoding of two labels (purple and yellow) in two eight-level neurons, while **F** shows the three-dimensional extension. The corresponding representational complexities $C$ are denoted in the bottom left corner.

robust estimation of the PID atoms, as small deviations in the empirical probabilities don't incur drastic changes in the representational complexity. Another less important property is differentiability as a function of the joint probability distribution $p(T, S)$. The $I^{\text{sx}}$ measure utilized in this paper has both properties (Makkeh et al., 2021).

Finally there are two intuitive bounds for the representational complexity. The representational complexity is lower than or equal to the maximum number of neurons and higher than or equal to one. The first bound is attained if one needs the whole layer to extract any the task-relevant information, while in the latter case, all information can be extracted from single neurons.

**Example applications of representational complexity** To become familiar with the intuitive interpretation of representational complexity, we demonstrate its use with simple information encodings in small toy examples. Consider a categorical random variable with a finite number of distinct classes, which are labelled by integers and occur with the same probability. How does representational complexity differ between different representations of this information in multiple binary neurons?

First, imagine four distinct values are represented sparsely across four neurons (Figure 2.A). As for all such *one-hot* encodings, the representational complexity of this encoding is equal to one (for a formal proof, refer to Appendix A.2.2). This is intuitively clear: For each realization, you only need access to the one neuron that is equal to 1 to fully determine the correct label.

Using the same number of neurons, we can also encode the same information in a more complex way. For instance, take pairs of neurons that redundantly represent digits of a binary representation of the label index (Figure 2.B). In this case, there is no longer a single neuron that contains the full information about the target, which is reflected by an increase of the representational complexity to $C = 1.21$.

Using all of the coding capacity of the four neurons from the example before by encoding 16 distinct states in a binary code (Figure 2.C), the representational complexity increases further: $C$ reaches a value of 1.67

as more information has to be encoded synergistically between the four neurons. The reason why $C$ is not equal to 4, despite the fact that the correct label can only be isolated with access to all four neurons, is that *parts* of the information, e.g., the parity of the label number, can be extracted from fewer than all sources, reflecting the average nature of $C$. Conversely, if we now again expand the 16 realizations to 16 neurons in a one-hot encoding (Figure 2.D), we revert back to a representational complexity of just $C = 1$. From this, we gain the intuitive realization that the closer one gets to the channel capacity, the more one is forced to encode some of the information in more complex, higher-synergy terms.

However, even small amounts of information can be encoded in a highly synergistic manner. Consider the more realistic case of two discrete artificial neurons with eight activation levels each, with the target value being the exclusive disjunction (XOR) of the thresholded neurons' activations (Figure 2.E). Despite encoding just a single bit of information, the representational complexity assumes a value of $C = 1.89$. This value is close to the theoretical maximum value of $C = n = 2$, with the small difference being due to the nature of the $I_\cap^{\mathrm{sx}}$ redundancy measure. Extending this task to the parity of three thresholded neurons (Figure 2.F), the representational complexity attains a value of $C = 2.83$, similarly close to its maximum of $C = n = 3$.

## 4 Application to deep neural networks

To exemplify the utility of our measure, we show how it can be applied to analyze the hidden layer representations of deep neural network classifiers solving the well-established MNIST (LeCun et al., 1998) handwritten digit recognition and CIFAR10 (Krizhevsky, 2009) image classification tasks. For the former, we employ a small fully-connected feed-forward network of which we analyze the last four layers before the output layer while for the latter we employ a more sophisticated convolutional network architecture and analyze the final small fully-connected layers.

The MNIST network consists of seven fully-connected layers, starting with a vector of the 784 grayscale pixel values of the image input, tapering down to only five neurons in layers $L_3$, $L_4$ and $L_5$ and culminating in a ten neuron *one-hot* output vector, in which each neuron represents one of the ten possible digits (Figure 3.A). The structure has been chosen such that the three successive five-neuron layers can be analyzed in full, as it is practically infeasible at present to compute the full PID for more than five sources because of the fast-growing number of PID atoms.

In order to limit the channel capacity to be able to observe non-trivial information dynamics (Goldfeld et al., 2019), all layers are discretized to eight or four quantization levels (3 or 2 bits) per neuron during both training and analysis (Appendix A.3). The networks are trained using stochastic gradient descent for $10^5$ training epochs and reach, with eight quantization levels, an average accuracy of $99.9(1)\,\%$ on the train and $95.1(4)\,\%$ on the test set for 20 runs with unique random weight initializations. Details about quantization schemes and forward-stochastic backprop algorithm used in training can be found in Appendix A.3.

The larger CIFAR10 network comprises three convolutional layers with max pooling and ReLU activation functions. The outputs of the last convolutional layer are flattened and fed into a fully-connected feed-forward section of the network employing tanh activation functions (Appendix A.3). Due to the conceptual difficulties with quantizing functions with semi-infinite ranges we applied the quantization with 8 levels only to the last layers with tanh activation function, which are also the ones analyzed later. For 10 random weight initializations, the CIFAR10 network achieves an average accuracy of $99.90(4)\,\%$ on the train and $68.9(7)\,\%$ on the test set after $10^4$ epochs of stochastic gradient descent training.

Except for when explicitly specified otherwise, all analyses have been done on the train dataset. The rationale for this is twofold: Firstly, since the train set samples drive the learning process, an analysis on the train dataset samples allows more insights into what information-theoretic representation the backpropagation algorithm implicitly pursues. Secondly, the train dataset - being defined as the set of samples the network is trained on - is trivially finite in size and can be analyzed in full. The test dataset, on the other hand, may be interpreted as merely a finite-sized sample (e.g. 60.000 handwritten digits for MNIST) of a potentially infinite generative process (e.g. the process of handwriting and formatting digits), which means that potential biases of the sampling and estimation need to be kept in mind. A comparison between results computed on the train and test datasets can be found in Section 4.5.

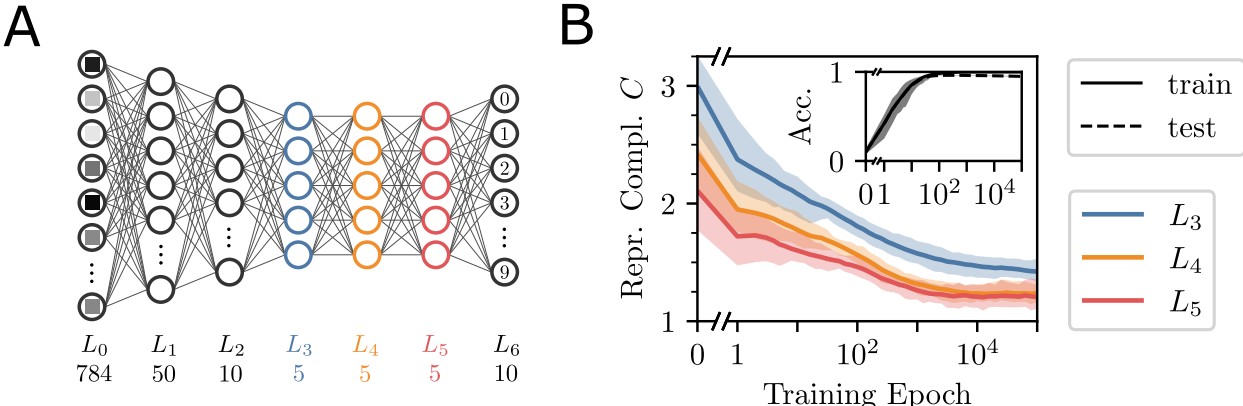

Figure 3: **Representational complexity of hidden layers with respect to the label decreases over training and throughout successive hidden layers. A** The MNIST classifier network consists of seven fully-connected layers with tanh activation functions for all but the last layer, which is equipped a softmax activation function. **B** The representational complexity of the three small hidden layers $L_3$, $L_4$ and $L_5$ are computed from their PID atoms on the train data set. The solid lines show the average of 20 randomly initialized runs, the shaded areas contain 95% of the data points. The inset shows train and test set accuracy.

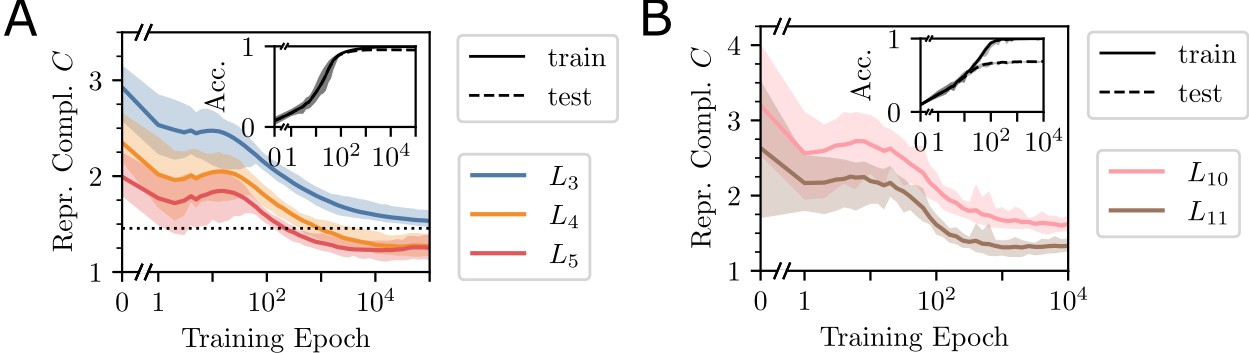

Figure 4: **The observed decrease in representational complexity appears to be a robust trend reproducible on different output encodings and on larger convolutional networks solving CIFAR10. A** The MNIST network with a binary output encoding shows a brief increase in representational complexity after about 10 epochs of training before decreasing. The representational complexity of the binary output layer for 10 equiprobable classes has been numerically determined to be $C = 1.46$ and is indicated by a dotted line. **B** The representational complexity of the small fully-connected layers of the CIFAR10 network shows a similar small increase in representational complexity in the beginning before converging to a low value close to the theoretical minimum of $C = 1$.

## 4.1 Representational complexity in small layers

For layers with up to five variables, the PID can be computed in full. For the MNIST network with eight quantization levels, analyzing the mutual information of the hidden layers $L_2$, $L_3$ and $L_4$ with the ground-truth label allows to track the representational complexity both over training and through the successive layers (Figure 3.B). We find that both with increasing training epoch and layer index, the representational complexity $C$ decreases.

This decrease in representational complexity appears to be a robust trend observable in networks independent of the encoding enforced on the output layer, chosen task and network architecture. While a neural network seems to approach the output complexity in the case of a one-hot output representation, for which $C_{\text{one-hot}} = 1$ (proven in Appendix A.2.2), the representational complexity of the hidden layers decreases below that of the

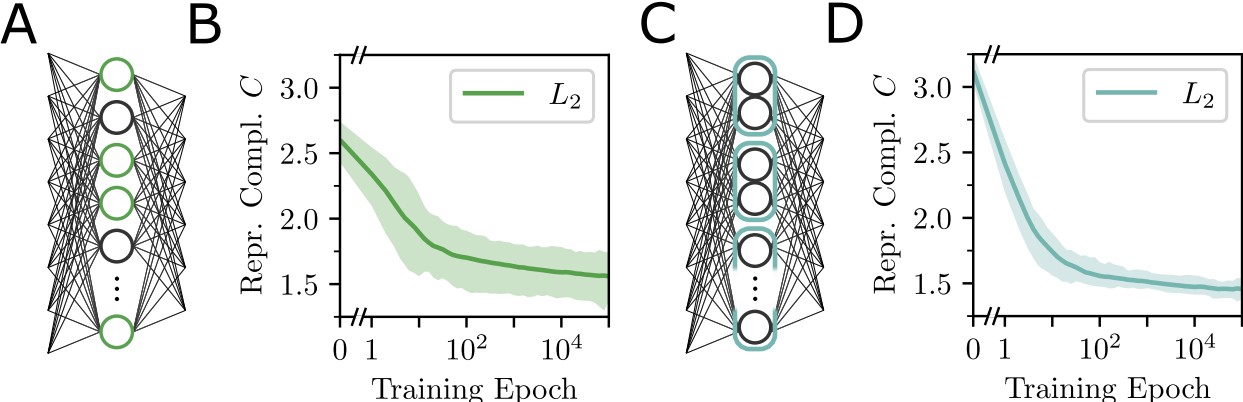

Figure 5: **Representational complexity of subsampled and coarse-grained neurons decreases over training in the MNIST network with four quantization levels. A** Five neurons are randomly selected from layer $L_2$ of the MNIST network to form the random variables for the subsampling analysis. **B** For 26 random choices of source variables each on 20 training runs, the subsampled representational complexity shows a decrease over the training phase, but with high variance. **C** Five coarse-grained random variables are constructed from pairs of neurons of the second hidden layer. **D** The coarse-grained representational complexity averaged over 20 training runs is seen to decrease with increasing training epoch.

targeted output representation in the case of a binary label encoding (Figure 4.A). In the binary encoding case, however, an intermediate increase in representational complexity is observable, which might be attributable to an early restructuring of the representation.

Likewise, for the five-neuron layers $L_{10}$ and $L_{11}$ of the CIFAR10 network, one observes a plateau of the representational complexity until about epoch 10 which is followed by a subsequent decrease (Figure 4.B). Again, $C$ is lower for the layer later in the network.

## 4.2 Representational complexity in larger layers

Typical production neural networks have layers which are much wider than five neurons (e.g., He et al., 2016; Krizhevsky et al., 2012). Since the compute required for a full PID for six or more source variables is prohibitive because of the rapidly increasing number of atoms, in order to be able to apply the tool of PID in general and representational complexity in particular to wider layers, one needs to devise procedures to reduce the number of random variables to analyze. In this section, we present two complementary approaches to make representational complexity applicable to moderately wider layers, namely subsampling and coarse-graining, and show the latter to be the more theoretically sound approach.

**Subsampling**  A straightforward approach for reducing the number of random variables, which has been employed in previous works (e.g., Tax et al., 2017), is to *subsample* only $\hat{n}$ neurons from a layer to use as PID sources. By randomly selecting five neurons from the second layer of the MNIST network with four quantization levels, we again observe a decrease of the representational complexity of the hidden layer with respect to the label over the training phase (Figure 5.A, B), albeit with a larger amount of variability.

However, this approach suffers from fundamental conceptual flaws. By choosing only $\hat{n}$ source variables $\boldsymbol{S_a} = (S_{a_1}, \dots, S_{a_{\hat{n}}})$ and decomposing their mutual information $I(T : \boldsymbol{S_a})$ with the target variable $T$, only atoms with a degree of synergy of less than or equal to $\hat{n}$ can be quantified, meaning that any higher interaction than $\hat{n}$ will be overlooked, resulting in a potential underestimation of the true representational complexity of all $n$ sources. At the same time, pieces of information that appear to only be obtainable synergistically within one subset $\boldsymbol{S_a}$ of sources may very well be redundant with a single source $S_i \notin \boldsymbol{S_a}$, thus leading to potential overestimation of representational complexity. For these reasons, no bound on the true representational complexity can be established from subsampling and we find subsampling to be an unsuitable approach for overcoming the scaling difficulties of PID.

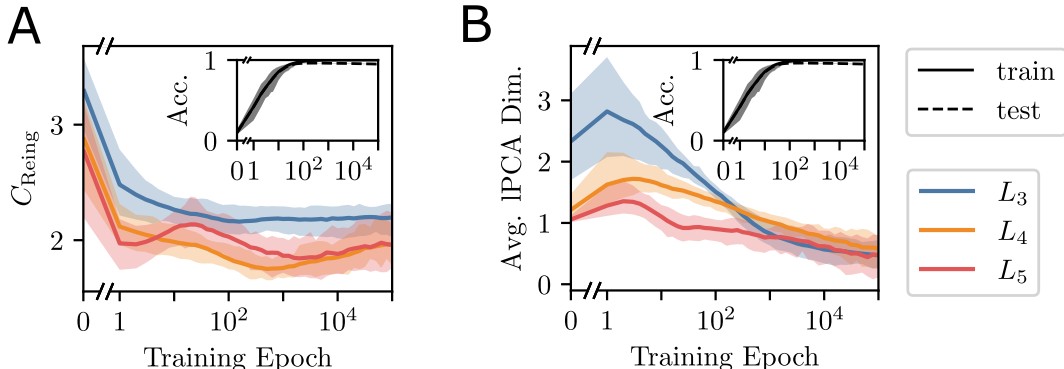

Figure 6: **Comparisons with alternative approaches shows decreasing trend but no ordering through successive hidden layers. (A)** The *Reing Representational Complexity* $C_{\text{Reing}}$ (Equation (9)) decreases only in the first epochs of training and seems to fluctuate at a relatively high level. **(B)** The *local intrinsic dimension* of the hidden layer activations of the MNIST classifier network based on the Fukunaga-Olsen algorithm (Fukunaga & Olsen, 1971) shows similar trends as representational complexity but increases at the beginning of training. For the intrinsic dimension estimation, the number of neighbors was set to 100, and the alphaFO parameter was set to 0.05, both being fairly common settings. **(A) and (B)** Both quantities have been computed for the same three hidden layers of the network as the results shown in Figure 3.

**Coarse-graining**  A complementary approach to reduce the number of variables is to combine multiple neurons, forming fewer higher-dimensional random variables; this procedure will be referred to as *coarse-graining*. This approach appears more theoretically sound because one can show that the actual representational complexity can be bounded up to a factor by the coarse graining representational complexity. In Appendix A.2.1, we prove that if $d$ neurons $S_i$ each are combined to random variables $\tilde{S}_j$, the true representational complexity of the layer is bounded from below by the coarse-grained representational complexity computed from $\tilde{\boldsymbol{S}} = \{\tilde{S}_1, \ldots, \tilde{S}_{n/d}\}$, while being simultaneously bounded from above by $d$ times this value, i.e.,

$$C(T : \tilde{\boldsymbol{S}}) \leq C(T : \boldsymbol{S}) \leq d\, C(T : \tilde{\boldsymbol{S}}). \tag{6}$$

In our example network with four quantization levels, the representational complexity computed from randomly assigned neuron pairs in the second hidden layer consisting of ten neurons, exhibits a decreasing pattern over training that is highly similar to that of the representational complexity computed with individual neurons as sources in layers $L_3$ to $L_5$ (Figure 5.C, D).

### 4.3   Comparison to Neural Information Decomposition by Reing et al.

In their 2021 paper, Reing et al. introduced an alternate way to decompose the mutual information between a target variable $T$ and multiple source variables $\boldsymbol{S}$ into what they term *Directed Local Differences* $C_T(k-1||k)$, such that

$$I(T : \boldsymbol{S}) = \sum_{k=1}^{n} C_T(k-1||k), \tag{7}$$

where $n$ is the number of sources. These terms, defined as

$$C_T(k-1||k) = \frac{1}{\binom{n}{k}} \sum_{|\boldsymbol{a}|=k} H(T|\boldsymbol{S_a}) - \frac{1}{\binom{n}{k-1}} \sum_{|\boldsymbol{a}|=k-1} H(T|\boldsymbol{S_a}) \tag{8}$$

are proposed by the authors as "a measure of information at order $k$ in the source variables $\boldsymbol{S}$ about a target variable $T$" (Reing et al., 2021) and are conceptually close analogs to the sum of PID atoms of a fixed degree of synergy. Nevertheless, due to their definition depending on the conditional entropy of the target

(in particular fractional sums of $H(T \mid \boldsymbol{S_a})$), these quantities cannot be constructed from the PID atoms. This means that a theoretical comparison is out of reach for this paper, whereas an empirical comparison between the two approaches might still be enlightening. Comparing the sums of PID atoms of a fixed degree of synergy to the directed differences for the MNIST network reveals that the sums of PID atoms attribute more of the information to only single neurons by the end of training, while the directed differences imply that higher-order interactions stay relevant to a higher degree (Figure 9).

In close analogy to Equation (5), one can utilize the directed differences $C_T(k-1||k)$ to introduce a measure of *Reing Representational Complexity* as

$$C_{\text{Reing}} = \frac{1}{I(T:\boldsymbol{S})} \sum_{k=1}^{n} C_Y(k-1||k) \; k \tag{9}$$

that captures the average order $k$ of information between the target $T$ and multiple sources $\boldsymbol{S}$.

While we empirically observe this Reing representational complexity to decrease in the earlier epochs of training, it does not decrease monotonically, stays higher than the PID based representational complexity and does not decrease throughout successive hidden layers (Figure 6A). While there is no a priori reason to expect a convergence to a particular value of complexity, we find the general trend of decreasing complexity over training and the ordering of $C$ for successive hidden layers – the latter of which is absent for $C_{\text{Reing}}$ – to be intuitive findings which might reflect the networks tendency to encode the relevant information in more and more simple terms.

These results show that representational complexity leads to different values while also arguably being the more well-founded measure compared to the directed differences, because it leverages the more expressive PID framework compared to Shannon mutual information. On the other hand, the approach using directed differences has the advantage of being more easily applicable a larger number of source variables.

### 4.4 Comparison to Local Intrinsic Dimension

Additionally to the comparison with the approach of Reing et al. (2021) we want to provide an independent baseline comparison with an established non-information theoretic measure. For this empirical comparison, we chose the local intrinsic dimensionality of the manifold of discretized layer activation patterns as described by Fukunaga & Olsen (1971). This method estimates the number of dimensions based on the largest singular values of the covariance matrix of the data, and is implemented in the scikit-dimension package (Bac et al., 2021). The reason we chose this method out of a multitude of potential candidates is that it tries to estimate the minimum number of parameters that are necessary to locally describe the data, which is similar to our approach. Taking the average over the dimensions estimated for each sample, however, is quite different from taking the average over PID atoms. Additionally, the intrinsic dimension is less dependent on the orientation of the manifold with respect to the individual neuron dimensions. However, the most important conceptual difference to our approach is that the intrinsic dimensionality is not immediately related to the label since it takes into account only the internal representations of the layers.

For the MNIST dataset, we see empirically that both the local intrinsic dimension and the representational complexity decrease with training, but the intrinsic dimension shows no consistent order throughout successive hidden layers (Figure 6B). The two measures seem to be related, however the lack of a consistent decrease over layers also points at a potential benefit of our approach: being target-oriented towards the label allows for a focus on the more relevant aspects of the representation in terms of solving the task.

### 4.5 Representational Complexity for Train and Test datasets

To further illustrate the behaviour and possible usecases of representational complexity, we next compare the representational complexity computed on the train dataset with results computed on the test dataset. The train dataset is the default choice for computing the representational complexity, since it defines the task that the networks actually optimize their representations for (see also Section 4 on page 9). In Figure 11, we compare the results on the train and test datasets for averages over multiple runs for the MNIST dataset, and we provide additionally the results for individual training runs in Figure 12. Generally, we find a consistent

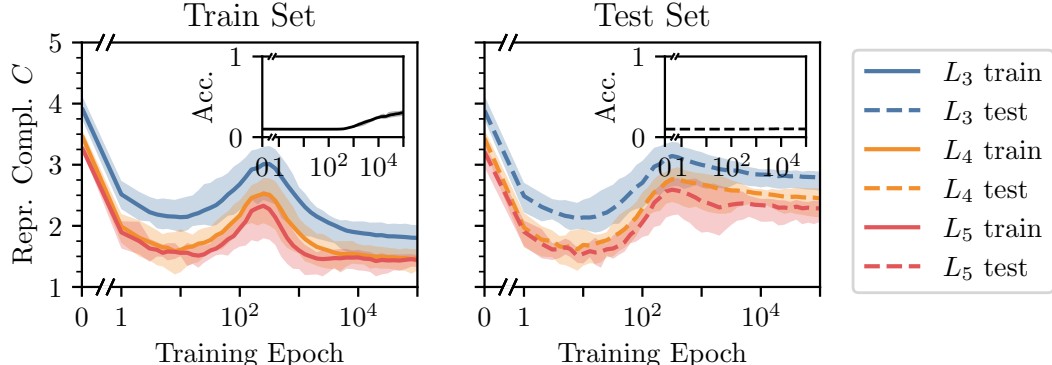

Figure 7: **When trained on an MNIST dataset with shuffled labels, representational complexity shows a different behaviour for train and test datasets.** Both figures show the representational complexity for the three hidden layers L3, L4 and L5, using the architecture illustrated in Figure 3A, but trained on a dataset with a random new assignment of labels to MNIST dataset images. Coinciding with the rise in training accuracy, the representational complexity for both train and test datasets reaches a pronounced peak around epoch 200. The representational complexity on the train dataset decreases towards the end of training. However, for the test dataset (right, also with shuffled labels) it remains substantially higher after this peak. For improved visibility of this phenomenon we combined the averages from both figures into a single diagram in Figure 10 in Appendix A.4.

and robust trend of higher representational complexity for the test dataset compared to that for the train dataset towards later stages of training.

In the extreme case of forcing the network to predict randomly assigned, fixed labels (by memorizing them via overfitting, see Figure 7) this increased representational complexity on the test dataset starts to appear with rising accuracy of the network and the gap between representational complexity on the train and test dataset is particularly wide.

While conclusive statements cannot be drawn from these empirical findings alone, using representational complexity for investigating the phenomenon of overfitting from a novel perspective appears as an enticing topic for further research.

## 5  Discussion

In this work, we introduced representational complexity as a measure of sparsity of task-relevant information in neural networks. We first explained how a principled and meaningful application of information theory in neural networks is possible by using quantized activation values in both training and evaluation. Subsequently, we derived representational complexity from partial information decomposition and show it to be the theoretically sound answer to the question of how many neurons need to be observed simultaneously in order to access an average piece of information. For larger layers, we presented subsampling and coarse-graining procedures. We discussed issues with the former approach, while deriving bounds from results of the latter approach.

In small quantized deep neural networks, we find representational complexity to decrease both over training and through successive layers. We empirically find the reduction to be a robust result which can not only be observed when analyzing small layers neuron-wise, but also when using subsampling or coarse-graining on larger layers. Furthermore, we provide first evidence that the final values the representational complexity converges to do not depend on the representational complexity of the chosen output representation. We hypothesize that the dynamics of representational complexity in early stages of training might indicate an early restructuring of representations for the binary output encoding and the more complex CIFAR10 task.

A comparison to the related information-theoretic tools introduced by Reing et al. (2021) shows that by focusing more on computational tractability on larger networks, Reing et al.'s measures can only partly reproduce our finding of a decreasing representational complexity throughout successive hidden layers. Similarly, an analysis based on local intrinsic dimensions (Fukunaga & Olsen, 1971) shows that while this dimension does indeed decrease over training, it does in practice not necessarily decrease with successive hidden layers, which is intuitively what we would expect for the complexity of the relevant part of the representation. This points to an important feature of our approach that many other comparable measures lack, namely the possibility to specify a target random variable. Thereby one is able to study a specific aspect of the representation, for example the degree of synergy of the task-relevant mutual information with the label. Discovering the precise differences between these related approaches on a theoretical ground, however, remains a promising avenue for future research.

Interestingly, towards the end of training we observe a consistent difference between the representational complexity computed on the train and the test dataset, pointing at a difference in the corresponding representation (see Section 4.5). In the extreme case of a network overfitting on randomly shuffled labels for the MNIST dataset (Figure 10) we report a substantial gap between the representational complexity on the train and test datasets. We believe that subsequent research studying this behaviour explicitly could result in new insights into the phenomenon of overfitting.

Our results also indicate some problems with previous approaches that apply PID by subsampling only pairs of sources (e.g. Tax et al. (2017)). Since the representational complexity is high in the early stages of training, high-order interactions between neurons appear to play a major role. These, however, go unnoticed in pairwise approaches. Representational complexity thus not only allows for assessing whether and when it is justified to only look at lower-order interactions, but can also be used to quantify how much of the higher-order interactions are disregarded.

**Limitations**  Scaling our approach to larger networks remains a challenge. To mitigate this problem, we introduced subsampling and course-graining procedures, which allow the application of representational complexity to moderately larger networks. However, while subsampling suffers from conceptual flaws, the proven attainable bounds on coarse graining become impractically loose for large networks. For typical production networks we suggest two approaches: Firstly, results found on small toy networks may be generalizable to larger networks, e.g., by inductive proofs, and secondly, estimation of representational complexity could become feasible by finding a way to compute it without computing all PID atoms beforehand. The latter can be achieved by employing a PID based on synergy instead of redundancy, for example by extending on ideas of Rosas et al. (2020).

Furthermore, our analysis methods are currently restricted to intrinsically discrete systems, as $I_\cap^{\mathrm{sx}}$ was originally defined for discrete variables only (Makkeh et al., 2021). However, in the meantime, a continuous generalization of $I_\cap^{\mathrm{sx}}$ has been proven to exist and to be measure-theoretically well-defined (Schick-Poland et al., 2021). Once an efficient estimator for this generalized measure is available, this will make it possible to analyze also continuous systems in which the total mutual information is inherently restricted to a finite value, e.g., by some form of noise in the system.

**Conclusion and outlook**  We have here shown how to use the representational complexity to gain insight into the structure of the internal representations as an average across all source and target realizations. Due to the local nature of $I_\cap^{\mathrm{sx}}$, this analysis can also be further broken down into the representational complexity for individual target realizations, i.e., class labels. Such an analysis may reveal for example that some classes are linked to representations of high complexity while others are not, or that some classes are represented with low complexity earlier than others during training, while yet others fail to reach a low complexity. Complementary, we have shown that a more detailed analysis could also be achieved by, instead of averaging over the different degrees of synergy, splitting up the total mutual information into "backbone atoms" (Rosas et al., 2020) which combine all atoms of the same degree of synergy $m$ and investigating how they evolve over training individually. In addition, once a proper measure for Synergy based-PID is proposed, one could use that fact that these Backbone atoms, as demonstrated by Rosas et al. (2020), can be computed individually which will allow for computing the representational complexity for significantly larger layers.

Some theoretical properties of representational complexity have yet to be uncovered. For instance, a lower bound to $C$ might be derived from the notion that closer to channel capacity, one is forced to represent some information in higher synergy terms.

Given the PID atoms, a whole suite of other interesting and easily interpretable quantities can be devised. One enticing candidate is the average multiplicity of information, defined as $M = (1/I(T : \boldsymbol{S})) \sum_\alpha \Pi(\alpha) |\alpha|$. The quantity reflects the average number of times a piece of information is represented redundantly and thus appears to be a promising candidate to serve as a complementary summary statistic to representational complexity, which relates to the synergy of the information encoding.

In future research, one may also choose the PID source and target variables differently. While in this work we focused on analyzing how the information about the classification label is encoded in the network, a similar analysis for the information of the hidden layers about the whole input samples might reveal differences in the representation of task-relevant and task-irrelevant information.

More generally, we promote representational complexity as a principled novel tool for general complex systems in which a group of equivalent variables jointly holds information about a target variable. New possible applications include both other artificial network architectures such as recurrent networks, but also biological or ecological systems. Being derived from first principles in information theory and partial information decomposition, representational complexity provides a clear and intuitive interpretation and the suggested subsampling and coarse-graining procedures make it applicable to a wide variety of questions.

## Code availability

To analyze artificial neural networks using information-theoretic tools, we developed the *nninfo* python package containing scripts to reproduce the main results of this paper. The code is available on GitHub under the URL `https://github.com/Priesemann-Group/nninfo`.

## Acknowledgements

We would like to thank Alexander Ecker for fruitful discussions and feedback on this paper. We would also like to thank Valentin Neuhaus, Lucas Rudelt, Marcel Graetz and the rest of the Priesemann group for their valuable comments and feedback. MW and AM are employed at the Göttingen Campus Institute for Dynamics of Biological Networks (CIDBN) funded by the Volkswagen Stiftung. DE and MW were supported by a funding from the Ministry for Science and Education of Lower Saxony and the Volkswagen Foundation through the "Niedersächsisches Vorab" under the program "Big Data in den Lebenswissenschaften" – project "Deep learning techniques for association studies of transcriptome and systems dynamics in tissue morphogenesis". AS and VP received funding from the Deutsche Forschungsgemeinschaft (DFG, German Research Foundation) under Germany's Excellence Strategy - EXC 2067/1 - 390729940.

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

# A  Appendix

This appendix starts by providing more background on the PID problem in Appendix A.1.1 by identifying the PID atoms by their parthood distributions as derived by Gutknecht et al. (2021) from first principles of mereology. Then we explain the equivalence of the parthood distribution and the antichains that was established by Gutknecht et al. (2021). Finally we describe in Appendix A.1.3 the lattice structure that allows computing the amount of information in the PID atoms upon the introduction of a redundancy measure as shown by the seminal work of Williams & Beer (2010).

In the subsequent two sections of the appendix we provide our novel proofs of the bounds on course graining (in Appendix A.2.1) and the proof of the representational complexity of the one-hot representation being one for any number of labels (in Appendix A.2.2).

Finally, we include a description of the implementation and the quantization schemes in Appendix A.3 and show additional results in Appendix A.4.

## A.1  Additional background

### A.1.1  Parthood distributions characterize the PID atoms

Gutknecht et al. (2021) recently showed that the PID structure can be built – independently of the redundancy concept and a specific measure – from first principles of mereology, i.e., *part-whole relationships.* This is because PID seeks to find the parts of information that constitute the whole information $I(T : S_1, \ldots, S_n)$ and are characterized by which collection of sources they are part of. In what follows we give a brief introduction to the PID structure using the mereology approach as has been developed by (Gutknecht et al., 2021, Section 2).

Through the lens of mereology, a PID atom is that part of the total information about the target that is provided by specific collections of source variables but not others. Thus, each atom is associated with a certain set of collections of sources $\boldsymbol{S_{a_1}}, \ldots, \boldsymbol{S_{a_m}} \in \mathcal{P}(\boldsymbol{S})$. The corresponding PID atom is then defined as the information contribution that can be obtained if and only if any of the collections is accessed.

For instance, in the case of two sources, consider all collections of the source variables, i.e., the powerset $\mathcal{P}(\boldsymbol{S}) = \{\emptyset, \{S_1\}, \{S_2\}, \{S_1, S_2\}\}$. Now by looking at what collections of sources the PID atoms can be a part of, Gutknecht et al. (2021) postulates two straightforward principles: (i) no information atom can be part of the empty set as it represents no source variable and (ii) all the information atoms are part of the full collection ($\{S_1, S_2\}$). This leaves four possibilities to combine the collections: Atoms can be either be a part of (a) both $\{S_1\}$ and $\{S_2\}$ or (b) only $\{S_1\}$ or (c) only $\{S_2\}$ or (d) only $\{S_1, S_2\}$ (Table 1).

| Atom | $\emptyset$ | $\{S_1\}$ | $\{S_2\}$ | $\{S_1, S_2\}$ |
|---|---|---|---|---|
| a) $\Pi(\{\{S_1\}, \{S_2\}\})$ | **0** | 1 | 1 | **1** |
| b) $\Pi(\{\{S_1\}\})$ | **0** | 1 | 0 | **1** |
| c) $\Pi(\{\{S_2\}\})$ | **0** | 0 | 1 | **1** |
| d) $\Pi(\{\{S_1, S_2\}\})$ | **0** | 0 | 0 | **1** |

Table 1: **Parthood distributions for two sources.** The value "0" stands for not being a part of the collection's information about $T$, as opposed to "1". "**0**" and "**1**" represent principle (i) and (ii), respectively. "0" and "1" represent the four possible choices of part-whole relationships to $\{S_1\}$ and $\{S_2\}$ realizing the four PID atoms. Each atom is annotated with its combination of collections of sources excluding supersets, e.g., the atom corresponding to possibility (a) is annotated with the collections $\{\{S_1\}, \{S_2\}\}$ as shorthand of $\{\{S_1\}, \{S_2\}, \{S_1, S_2\}\}$ and so on.

The first atom in the table is $\Pi(\{\{S_1\}, \{S_2\}\})$ that is part of $S_1$ and $S_2$. This atom represents what is redundant to $S_1$ and $S_2$ as it is part of both variables. The second and third atoms are exclusively parts of either $S_1$ and $S_2$, thereby they are unique to either variable. These two atoms represent what is unique to $S_1$ and $S_2$ respectively. Finally, the atom $\Pi(\{\{S_1, S_2\}\})$ is only part of the set $\{S_1, S_2\}$ but none of the sources

individually. This constitutes an information contribution that cannot be realised by any of the two variables unless they are considered jointly, capturing the synergistic information.

This PID, indeed, serves its aim to partition $I(T : (S_j)_{j \in J})$ for any number of sources $|J|$ into the realized atoms via the part-whole relationships. For example, in the case of two sources, we arrive at the following partition (Gutknecht et al., 2021; Williams & Beer, 2010):

$$I(T : S_1, S_2) = \Pi(\{\{S_1\}, \{S_2\}\}) + \Pi(\{\{S_1\}\}) + \Pi(\{\{S_2\}\}) + \Pi(\{\{S_1, S_2\}\}) ,$$
$$I(T : S_1) = \Pi(\{\{S_1\}, \{S_2\}\}) + \Pi(\{\{S_1\}\}) ,$$
$$I(T : S_2) = \Pi(\{\{S_1\}, \{S_2\}\}) + \Pi(\{\{S_2\}\}).$$

Each atom is represented with a string of zeros and ones referred to as a *parthood distribution* (Table 1). Gutknecht et al. (2021) then uses these parthood distributions to build the structure of PID for any number of sources. We have already seen that these parthood distributions follow two principles: (i) none of the PID atoms are part of the $\emptyset$ (ii) any PID atom is part of the joint set of source variables. However, an additional principle arises from the nature of the information contributions and is vital when considering more than two sources: The principle states that whenever an atom is part of the information provided by a collection $S_a$, then it must be part of the information provided by *any collection containing $S_a$*.

Hence, each PID atom is associated with a parthood distribution that is formally a boolean function $\Phi : \mathcal{P}(\{1, \ldots, n\}) \to \{0, 1\}$ satisfying three principles: (i) *no information is in the empty set*, i.e. $\Phi(\emptyset) = 0$, (ii) *all information is in the full set*, i.e. $\Phi(\{1, \ldots, n\}) = 1$, and (iii) any *information that is provided by collection $a$ is provided by any of its super-collections*, which is equivalent to saying that $\Phi$ is monotone. An example of such association is present in Figure 8.

| Atom | $\{\emptyset\}$ | $\{1\}$ | $\{2\}$ | $\{3\}$ | $\{1,2\}$ | $\{1,3\}$ | $\{2,3\}$ | $\{1,2,3\}$ | Boolean function |
|---|---|---|---|---|---|---|---|---|---|
| $\Pi(\{\{1\}\})$ | 0 | *1* | 0 | 0 | 1 | 1 | 0 | 1 | $f_{\Pi(\{\{1\}\})}$ |
| $\Pi(\{\{1\}, \{2,3\}\})$ | 0 | *1* | 0 | 0 | 1 | 1 | *1* | 1 | $f_{\Pi(\{\{1\}, \{2,3\}\})}$ |

Figure 8: **Parthood distributions are isomorphic to PID atoms** In this figure, we use the shorthand notation by dropping the $S$'s and resorting only to the indices of source variables. The first atom $\Pi(\{1\})$ is the unique information of $S_1$. This information is part of $S_1$ (i.e. $\{1\}$) and by principle (iii) any superset of $\{1\}$. The boolean function $f_{\Pi(\{1\})}$ realized from the parthood distribution maps $\{1\}$ and all its supersets to 1 whereas the rest are mapped to zero. The second atom $\Pi(\{\{1\}, \{2,3\}\})$ is the information that is exclusively redundant between $S_1$ and the joint source variable of $S_2$ and $S_3$, but not redundant between $S_1$ and any strict subsets of $\{S_2, S_3\}$. This can be realized from the parthood distribution as the information is part of $\{1\}$ and by principle (iii) all its supersets, however, additionally this information is also part of $\{2, 3\}$. Finally this parthood distribution is packed into the monotone function boolean function $f_{\Pi(\{\{1\}, \{2,3\}\})}$.

Therefore, the number of PID atoms for $n$ sources is equal to the number of monotone boolean functions over $\mathcal{P}(\{1, \ldots, n\})$ minus 2 since the two monotone boolean functions $\Phi(a) = 0$ for all $a$ and $\Phi(a) = 1$ for all $a$ are excluded by principles (i) and (ii), respectively Gutknecht et al. (2021). This entails that the number of atoms grows super-exponentially as the number of monotone boolean functions, i.e., as the Dedekind numbers Beckurts & Dedekind (1897).

### A.1.2 Equivalence between parthood distributions and antichains

In this section, we explain the equivalence between the parthood distribution and antichains that has been demonstrated by Gutknecht et al. (2021, Section 2). An information atom $\Pi$ can be equivalently referenced by either its parthood distribution $\Phi$ as $\Pi(T : S_\Phi)$ or its corresponding antichain $\alpha$ as $\Pi(T : S_\alpha)$. This equivalence is given by the facts that (i) the parthood distributions $\Phi : \mathcal{P}(\{1, \ldots, n\}) \to \{0, 1\}$ are *monotonic*, i.e., they fulfill the relation $a \subseteq b \Rightarrow \Phi(a) \leq \Phi(b)$, and (ii) antichains are succinct representations of such

monotonic boolean functions. To gain a deeper insight into the equivalent descriptions of the atoms, we will thus address these two points one by one in the following.

Gutknecht et al. introduces the monotonicity of the parthood distribution as a constraint in its definition (Gutknecht et al., 2021, definition 2.1.). In fact, one can see that this monotonicity is in line with an essential property of mutual information: The mutual information $I(T : \boldsymbol{S_a})$ between the target $T$ and a subset of sources $\boldsymbol{S_a}$ is contained in the mutual information $I(T : \boldsymbol{S_b})$ whenever $\boldsymbol{b}$ is a superset of $\boldsymbol{a}$. This fact is a result of the chain rule of mutual information and the non-negativity of (conditional) mutual information (Cover & Thomas, 2006) $I(T : \boldsymbol{S_b}) = I(T : \boldsymbol{S_a}, \boldsymbol{S_{b \setminus a}}) = I(T : \boldsymbol{S_a}) + I(T : \boldsymbol{S_{b \setminus a}} | \boldsymbol{S_a}) \geq I(T : \boldsymbol{S_a})$ which implies that the correlation between $S_a$ and $T$ still persists in the correlation of $S_b$ and $T$ and in fact only additional correlation can emerge with added sources. Thus when thinking about atoms that decompose the mutual information, it is only natural to evoke the constraint that if an atom $\Pi$ is part of the mutual information $I(T : \boldsymbol{S_a})$, i.e., $\Phi(\boldsymbol{a}) = 1$, it also has to be part of $I(T : \boldsymbol{S_b})$, i.e., $\Phi(\boldsymbol{b}) = 1$ whenever $\boldsymbol{a} \subseteq \boldsymbol{b}$, which is exactly the property of monotonicity: $\boldsymbol{a} \subseteq \boldsymbol{b} \Rightarrow \Phi(\boldsymbol{a}) \leq \Phi(\boldsymbol{b})$.

So far we saw that parthood distributions are monotonic boolean functions and we will discuss how to represent them in a succinct way as antichains. To this end, note that a boolean function is uniquely defined by the set of inputs $f^{-1}[\{1\}] := \{\boldsymbol{a} \in \mathcal{P}(\{1, \dots, n\}) \mid f(\boldsymbol{a}) = 1\}$ that are mapped to "1". Given the constraint of monotonicity, this representation can be compressed even further: Any sets in $f^{-1}[\{1\}]$ which are supersets of other sets in $f^{-1}[\{1\}]$ need not be recorded, as these are forced to map to "1" by monotonicity. Thus, a monotonic boolean function can be represented by $\alpha \subseteq f^{-1}[\{1\}]$, constructed by removing all sets in $f^{-1}[\{1\}]$ which are supersets of others. The resulting set of sets $\alpha$ then contains only sets which are incomparable given the partial order of set inclusion, which are referred to in the literature as "antichains" (Williams & Beer, 2010).

Therefore, each PID $I(T : \boldsymbol{S_\Phi})$ atom identified with a parthood distribution $\Phi$ can be represented by the antichain representation of $\Phi$, which provides an equivalent labelling of the atom as $I(T : \boldsymbol{S_\alpha})$.

### A.1.3  The lattice structure of PID using redundant information

In this section, we will discuss the well-known redundancy lattice and how this lattice structure of PID (Williams & Beer, 2010) arises when using redundant information. First, recall that redundant information $I_\cap(T : \boldsymbol{S_\Phi})$ is the information that is part of all $\boldsymbol{S_a}$ for which $\Phi(\boldsymbol{a}) = 1$. This means that $I_\cap(T : \boldsymbol{S_\Phi})$ is part of all the mutual information terms $I(T : S_a)$ where $\Phi(\boldsymbol{a}) = 1$. This part-whole relationship can be exploited to formulate a partial ordering of the general redundancies $I_\cap(T : \boldsymbol{S_\Phi})$. We say that that a redundancy $I_\cap(T : \boldsymbol{S_\Phi})$ is contained in another redundancy $I_\cap(T : \boldsymbol{S_\Psi})$ if the redundancy $I_\cap(T : \boldsymbol{S_\Phi})$ is necessarily a part of all the mutual information that $I_\cap(T : \boldsymbol{S_\Psi})$ is part of. This implies that any part of the information contained in $I_\cap(T : \boldsymbol{S_\Phi})$ is also necessarily contained in $I_\cap(T : \boldsymbol{S_\Psi})$ which forms the basis of this ordering. This ordering of the redundant information $I_\cap(T : \boldsymbol{S_\Phi})$ can be expressed in terms of parthood distributions as (Gutknecht et al., 2021, Subsection 2.iii):

$$\Phi \sqsubseteq \Psi \Leftrightarrow (\Phi(\boldsymbol{a}) = 1 \rightarrow \Psi(\boldsymbol{a}) = 1 \text{ for any } \boldsymbol{a} \subseteq \{1, \dots, n\}). \tag{10}$$

Using the equivalence between parthood distributions and antichains described in Appendix A.1.2, this partial order can be similarly expressed in terms of antichains as (Williams & Beer, 2010)

$$\alpha \preceq \beta \Leftrightarrow \forall \boldsymbol{b} \in \beta \; \exists \boldsymbol{a} \in \alpha : a \subseteq b. \tag{11}$$

This partial ordering defines the algebraic structure of the "Redundancy Lattice". We can now define the PID atoms as the partial differences on this lattice, i.e., the atom $\Pi(T : \boldsymbol{S_\Phi})$ contains all the information that the redundancy $I_\cap(T :_\Phi)$ has *in excess of* the sum of all atoms below $\Phi$ on the lattice. Conversely, this allows the redundancies $I_\cap(T : \boldsymbol{S_\Phi})$ to be the sum of all the atoms that are identified by a parthood distribution that is less than or equal $\Phi$ since all of the information in these atoms is contained in $\Pi(T : \boldsymbol{S_\Phi})$ via the redundancy ordering. This can be equivalently expressed in terms of antichain (due to the aforementioned

ordering) yielding the system of equations discussed in the main text:

$$I_\cap(T : \boldsymbol{S}_\alpha) = \sum_{\beta \preceq \alpha} \Pi(T : \boldsymbol{S}_\beta). \tag{12}$$

This system of equations states that for any node $\alpha$ of the "Redundancy Lattice", there is a redundant information that is the sum of the atom $\Pi(T : \boldsymbol{S}_\beta)$ for any $\beta \preceq \alpha$. This system of equations can be inverted to compute the amount of information in each individual atom using the Moebius Inversion due to the distributive structure of the Redundancy lattice (Williams & Beer, 2010).

## A.2 Mathematical proofs

### A.2.1 Proof of bounds on representational complexity by coarse-graining

To make representational complexity applicable to settings with more source random variables, we propose coarse-graining, i.e., combining source variables to form fewer, but higher dimensional, "super variables", as a suitable procedure. As a first step, we clarify how the new coarse-grained variables are constructed from the original variables using a *coarse-grain mapping*.

**Definition A.1** (Coarse-Grain mapping). For $n, \tilde{n} \in \mathbb{N}_{>0}$ and $\tilde{n} < n$, an $n$-to-$\tilde{n}$ coarse-grain mapping $f : \{1, \ldots, n\} \to \{1, \ldots, \tilde{n}\}$ is a surjective function that maps variable indices to fewer coarse-grained variable indices.

We write the pre-image of the coarse-grain mapping for subsets of coarse-grained source indices $\tilde{\boldsymbol{a}} \subseteq \{1, \ldots, \tilde{n}\}$ as $f^{-1}[\tilde{\boldsymbol{a}}] = \{i \in \{1, \ldots, n\} | f(i) \in \tilde{\boldsymbol{a}}\}$.

Furthermore, we write sets of random variables indexed by the set of indices $\boldsymbol{a} \in \mathcal{P}(\{1, \ldots, n\})$ as $\boldsymbol{S}_{\boldsymbol{a}} = \{S_i | i \in \boldsymbol{a}\}$ and finally sets of sets of random variables indexed by sets of sets of indices $\alpha \in \mathcal{P}(\mathcal{P}(\{1, \ldots, n\}))$ as $\boldsymbol{S}_\alpha = \{\boldsymbol{S}_{\boldsymbol{a}} | \boldsymbol{a} \in \alpha\}$.

Using this notion of a coarse-grain mapping, we can define what a coarse-grained random variables is.

**Definition A.2** (Coarse-Grained Random Variable). Given a vector-valued random variable $\boldsymbol{S} = (S_1, \ldots, S_n)$ and an $n$-to-$\tilde{n}$ coarse-grain mapping $f$, the coarse-grained vector-valued random variable $\tilde{\boldsymbol{S}}$ is defined as $\tilde{\boldsymbol{S}} = (\tilde{\boldsymbol{S}}_1, \ldots, \tilde{\boldsymbol{S}}_{\tilde{n}})$, where the elements $\tilde{\boldsymbol{S}}_i = \boldsymbol{S}_{f^{-1}[\{i\}]} = \{S_k | f(k) = i\}$ are themselves vector-valued random variables, called coarse-grained variables, partitioning the $n$ original variables into $\tilde{n}$ variables.

**Example A.1.** Given four source variables $\boldsymbol{S} = (S_1, S_2, S_3, S_4)$, the mapping

$$f : \{1, 2, 3, 4\} \to \{1, 2\}, \; i \mapsto \begin{cases} 1, \; i \in \{1, 2\} \\ 2, \; i \in \{3, 4\} \end{cases}$$

defines the two coarse-grained random variables $\tilde{\boldsymbol{S}}_1 = (S_1, S_2)$ and $\tilde{\boldsymbol{S}}_2 = (S_3, S_4)$.

In general, a coarse-grain mapping can produce coarse-grained variables consisting of different numbers of original variables. An important special case, however, is that of the *uniform coarse-grain mapping*, which always maps $d$ source variables to one $d$-dimensional "super-variable".

**Definition A.3** (Uniform coarse-grain mapping). A uniform coarse-grain mapping of order $d \in \mathbb{N}_{>0}$ such that $d|n$ is given by $f : \{1, \ldots, n\} \to \{1, \ldots, n/d\}$, $i \mapsto \lfloor (i-1)/d \rfloor + 1$, where $\lfloor \cdot \rfloor$ refers to rounding down to the nearest integer.

Having defined the coarse-grained variables $\tilde{\boldsymbol{S}}$, the next question that arises is how the PID atoms of the original variables can be combined to form the coarse-grained PID atoms of $\tilde{\boldsymbol{S}}$.

**Theorem A.1** (Coarse-graining of PID atoms). *The PID atoms $\Pi(T : \tilde{\boldsymbol{S}}_{\tilde{\Phi}})$ of the coarse-grained source variable $\tilde{\boldsymbol{S}}$ are composed of the PID atoms $\Pi(T : \boldsymbol{S}_\Phi)$ of the original sources $\boldsymbol{S}$ as*

$$\Pi\left(T : \tilde{\boldsymbol{S}}_{\tilde{\Phi}}\right) = \sum_{\Phi : \Phi \circ f^{-1} = \tilde{\Phi}} \Pi\left(T : \boldsymbol{S}_\Phi\right). \tag{13}$$

*Proof.* A PID atom $\Pi\left(T : \boldsymbol{S}_\Phi\right)$ is a part of the coarse-grained PID atom $\Pi\left(T : \tilde{\boldsymbol{S}}_{\tilde{\Phi}}\right)$ exactly if it contributes to the same mutual information terms $I\left(T : \tilde{\boldsymbol{S}}_{\tilde{\boldsymbol{a}}}\right)$ with the coarse-grained variables $\tilde{\boldsymbol{S}}_{\tilde{\boldsymbol{a}}}$. Since the identity

$$I\left(T : \tilde{\boldsymbol{S}}_{\tilde{\boldsymbol{a}}}\right) = I\left(T : \boldsymbol{S}_{f^{-1}[\tilde{\boldsymbol{a}}]}\right)$$

follows readily from the definition of the coarse-grained variables, we find that if $\Pi\left(T : \boldsymbol{S}_\Phi\right)$ is part of $I\left(T : \boldsymbol{S}_{f^{-1}[\tilde{\boldsymbol{a}}]}\right)$, i.e., $\Phi(f^{-1}[\tilde{\boldsymbol{a}}]) = 1$, it is also part of $I\left(T : \tilde{\boldsymbol{S}}_{\tilde{\boldsymbol{a}}}\right)$ in the coarse-grained picture, and vice-versa. Thus, the parthood relations of $\Pi\left(T : \boldsymbol{S}_\Phi\right)$ with regards to the coarse-grained variables are captured by the parthood distribution $\tilde{\Phi} = \Phi \circ f^{-1}$, which leads to the conclusion that it must be part of the coarse-grained atom $\Pi\left(T : \tilde{\boldsymbol{S}}_{\tilde{\Phi}}\right)$. The coarse-grained atoms must now be the sums of all original atoms which contribute to the same coarse-grained mutual information terms; thus the theorem follows.

$\square$

This coarse-grained PID naturally gives rise to both a lower and an upper bound on the representational complexity of the original variables. As a first step, we show how representational complexity can be equally computed from the parthood distribution $\Phi$ of an atom.

**Lemma A.1** (Computing the degree of synergy from a parthood distribution)**.** *The degree of synergy of the atom indexed by the parthood distribution $\Phi$ is given by $m(\Phi) = \min_{\Phi(\boldsymbol{a})=1} |\boldsymbol{a}|$.*

*Proof.* The parthood distribution $\Phi$ corresponding to an antichain $\alpha$ is the boolean function that maps all $\boldsymbol{a} \in \alpha$ and all supersets thereof to one. This means, in addition to all sets $\boldsymbol{a} \in \alpha$, that $\Phi^{-1}[\{1\}]$, the fibre of 1 under $\Phi$, contains only sets $\boldsymbol{a}' \supset \boldsymbol{a}$, for which $|\boldsymbol{a}'| > |\boldsymbol{a}|$ and which thus have no influence on the minimum cardinality of sets:

$$m(\alpha) := \min_{\boldsymbol{a} \in \alpha} |\boldsymbol{a}| = \min_{\boldsymbol{a} \in \Phi^{-1}[\{1\}]} |\boldsymbol{a}| = \min_{\Phi(\boldsymbol{a})=1} |\boldsymbol{a}| =: m(\Phi) . \tag{14}$$

$\square$

The next step in our quest to prove bounds on the representational complexity is to prove bounds on the degree of synergy $m$.

**Lemma A.2.** *Let $\Phi$ and $\tilde{\Phi}$ refer to the parthood distributions of an atom and a coarse-grained atom, respectively. If $\Pi(T : \boldsymbol{S}_\Phi)$ is part of $\Pi(T : \tilde{\boldsymbol{S}}_{\tilde{\Phi}})$, then the degree of synergy $m(\Phi)$ is constrained by the degree of synergy $m(\tilde{\Phi})$ as $m(\tilde{\Phi}) \leq m(\Phi) \leq d\,m(\tilde{\Phi})$, where the upper bound holds for a uniform coarse-graining of order $d$.*

*Proof.* Let the atom $\Pi(T : \boldsymbol{S}_\Phi)$ be part of the coarse-grained atom $\Pi(T : \tilde{\boldsymbol{S}}_{\tilde{\Phi}})$. It follows from Equation (13) that $\tilde{\Phi} = \Phi \circ f^{-1}$ and therefore

$$m(\tilde{\Phi}) = \min_{\tilde{\Phi}(\tilde{\boldsymbol{a}})=1} |\tilde{\boldsymbol{a}}| = \min_{\Phi \circ f^{-1}(\tilde{\boldsymbol{a}})=1} |\tilde{\boldsymbol{a}}| = \min_{\Phi(\boldsymbol{a})=1} |f(\boldsymbol{a})| \leq \min_{\Phi(\boldsymbol{a})=1} |\boldsymbol{a}| = m(\Phi) ,$$

where the fact that $|f(\boldsymbol{a})| \leq |\boldsymbol{a}|$ has been used. Note similarly that for a uniform coarse-graining of order $d$, $d|f(\boldsymbol{a})| \geq |\boldsymbol{a}|$ holds, hence one finds a lower bound to $m(\tilde{\Phi})$ as

$$m(\tilde{\Phi}) = \min_{\Phi(\boldsymbol{a})=1} |f(\boldsymbol{a})| \geq \frac{1}{d} \min_{\Phi(\boldsymbol{a})=1} |\boldsymbol{a}| = \frac{1}{d} m(\Phi) .$$

$\square$

From the bounds on the synergistic degree, the bounds on the representational complexity - which is a weighted average of synergistic degrees - are straightforward to derive.

**Theorem A.2.** *The representational complexity of a vector of sources $\boldsymbol{S} = (S_1, \ldots, S_n)$ with respect to some target $T$ is bounded from below by the representational complexity of any coarse-graining $\tilde{\boldsymbol{S}}$*

$$C(T : \tilde{\boldsymbol{S}}) \leq C(T : \boldsymbol{S}). \tag{15}$$

*Proof.*

$$C(T : \tilde{\boldsymbol{S}}) = \frac{1}{I(T : \tilde{\boldsymbol{S}})} \sum_{\tilde{\Phi}} \Pi(T : \tilde{\boldsymbol{S}}_{\tilde{\Phi}}) \, m(\tilde{\Phi})$$

$$= \frac{1}{I(T : \tilde{\boldsymbol{S}})} \sum_{\tilde{\Phi}} \left( \sum_{\Phi : \Phi \circ f^{-1} = \tilde{\Phi}} \Pi(T : \boldsymbol{S}_{\Phi}) \right) m(\tilde{\Phi}) \qquad \text{(Theorem A.1)}$$

$$\leq \frac{1}{I(T : \tilde{\boldsymbol{S}})} \sum_{\tilde{\Phi}} \sum_{\Phi : \Phi \circ f^{-1} = \tilde{\Phi}} \Pi(T : \boldsymbol{S}_{\Phi}) \, m(\Phi) \qquad \text{(Lemma A.2)}$$

$$= \frac{1}{I(T : \boldsymbol{S})} \sum_{\Phi} \Pi(T : \boldsymbol{S}_{\Phi}) \, m(\Phi)$$

$$= C(T : \boldsymbol{S}).$$

$\square$

**Theorem A.3.** *The representational complexity of a vector of sources $\boldsymbol{S} = (S_1, \ldots, S_n)$ with respect to some target $T$ is bounded from above by $d$ times the representational complexity of a uniform coarse-graining of order $d$*

$$C(T : \boldsymbol{S}) \leq d \, C(T : \tilde{\boldsymbol{S}}). \tag{16}$$

*Proof.*

$$C(T : \tilde{\boldsymbol{S}}) = \frac{1}{I(T : \tilde{\boldsymbol{S}})} \sum_{\tilde{\Phi}} \Pi(T : \tilde{\boldsymbol{S}}_{\tilde{\Phi}}) \, m(\tilde{\Phi})$$

$$= \frac{1}{I(T : \tilde{\boldsymbol{S}})} \sum_{\tilde{\Phi}} \left( \sum_{\Phi : \Phi \circ f^{-1} = \tilde{\Phi}} \Pi(T : \boldsymbol{S}_{\Phi}) \right) m(\tilde{\Phi}) \qquad \text{(Theorem A.1)}$$

$$\geq \frac{1}{I(T : \tilde{\boldsymbol{S}})} \sum_{\tilde{\Phi}} \sum_{\Phi : \Phi \circ f^{-1} = \tilde{\Phi}} \Pi(T : \boldsymbol{S}_{\Phi}) \, m(\Phi)/d \qquad \text{(Lemma A.2)}$$

$$= \frac{1}{d \, I(T : \boldsymbol{S})} \sum_{\Phi} \Pi(T : \boldsymbol{S}_{\Phi}) \, m(\Phi)$$

$$= \frac{1}{d} C(T : \boldsymbol{S}).$$

$\square$

### A.2.2 Proof of representational complexity of one-hot encoding

In DNNs solving a classification task, the output labels are typically encoded in a "One-Hot Encoding", in which there are as many neurons as classes with only the neuron corresponding to the correct class being one while all others are zero. Here we prove that all such encodings have a representational complexity of $C = 1$.

**Definition A.4** (One-hot encoding). The one-hot encoding $\vec{Y}$ of a categorical variable $Y$ with finite ordered alphabet $A_Y = (y_1, y_2, \ldots, y_n)$ is the image of the bijective mapping

$$\vec{\cdot} : A_Y \to A_{\vec{Y}} \subset \{0, 1\}^n, y \mapsto \vec{y} = (\delta_{y, y_j})_j = (0, \ldots, \underbrace{0}_{j-1}, \underbrace{1}_{j}, \underbrace{0}_{j+1}, \ldots, 0),$$

where $\delta_{y,y_j}$ is the Kronecker Delta.

In what follows, we recall the concepts of local mutual information $i$, local $I_\cap^{\text{sx}}$ redundancy $i_\cap^{\text{sx}}$, and its additive decomposition into local informative redundancy $i_\cap^{\text{sx}+}$ and local misinformative redundancy $i_\cap^{\text{sx}-}$. These information functionals are needed in proving that any one-hot encoding has a representational complexity equals to one.

**Definition A.5** (Local information and their informative and misinformative parts)**.** Let $T$ be the target variable and $\boldsymbol{S}$ be the set of sources. Then, we have the following:

- The mutual information $I(T : \boldsymbol{S})$ is, in fact, the expected value of the local mutual information $i(t : \boldsymbol{s})$ as follows:

$$I(T : \boldsymbol{S}) := \sum_{t,\boldsymbol{s}} \mathbb{P}(\mathfrak{t} \cap \mathfrak{s}) \log_2 \frac{\mathbb{P}(\mathfrak{t} \cap \mathfrak{s})}{\mathbb{P}(\mathfrak{t})\,\mathbb{P}(\mathfrak{s})} = \mathbb{E}_{t,\boldsymbol{s}}\left[i(t : \boldsymbol{s})\right] .$$

- The local mutual information $i(t : \boldsymbol{s})$ can take negative values and so it is decomposed into non-negative informative $i^+(t : \boldsymbol{s})$ and misinformative $i^-(t : \boldsymbol{s})$ parts as follows:

$$i^+(t : \boldsymbol{s}) := \log_2 \frac{1}{\mathbb{P}(\mathfrak{s})} = \log_2 \frac{1}{p_{\boldsymbol{S}}(s)} ,$$

$$i^-(t : \boldsymbol{s}) := \log_2 \frac{\mathbb{P}(\mathfrak{t})}{\mathbb{P}(\mathfrak{t} \cap \mathfrak{s})} = \log_2 \frac{1}{p_{\boldsymbol{S}|T}(s \mid t)} .$$

- The $I_\cap^{\text{sx}}$ redundancy $I_\cap^{\text{sx}}(T : \boldsymbol{S}_\alpha)$ is in its turn the expected value of the local redundant information $i^{\text{sx}}(t : \boldsymbol{s}_\alpha)$ as follows:

$$I_\cap^{\text{sx}}(T : \boldsymbol{S}_\alpha) = \sum_{t,\boldsymbol{s}} \mathbb{P}(\mathfrak{t} \cap \mathfrak{s}) \log_2 \frac{\mathbb{P}(\mathfrak{t}) - \mathbb{P}\left(\mathfrak{t} \cap \bigcap_{\boldsymbol{a} \in \alpha} \bar{\mathfrak{s}}_{\boldsymbol{a}}\right)}{\mathbb{P}(\mathfrak{t})\left[1 - \mathbb{P}\left(\bigcap_{\boldsymbol{a} \in \alpha} \bar{\mathfrak{s}}_{\boldsymbol{a}}\right)\right]} = \mathbb{E}_{t,\boldsymbol{s}}\left[i_\cap^{\text{sx}}(t : \boldsymbol{s}_\alpha)\right] .$$

- The local $I_\cap^{\text{sx}}$ redundancy $i_\cap^{\text{sx}}(t : \boldsymbol{s}_\alpha)$ can also take negative values and so it is decomposed into nonnegative informative $i_\cap^{\text{sx}+}(t : \boldsymbol{s}_\alpha)$ and misinformative $i_\cap^{\text{sx}-}(t : \boldsymbol{s}_\alpha)$ parts as follows:

$$i_\cap^{\text{sx}+}(t : \boldsymbol{s}_\alpha) := \log_2 \frac{1}{1 - \mathbb{P}\left(\bigcap_{\boldsymbol{a} \in \alpha} \bar{\mathfrak{s}}_{\boldsymbol{a}}\right)} = \log_2 \frac{1}{p_{\boldsymbol{S}_\alpha}(s_\alpha)} ,$$

$$i_\cap^{\text{sx}-}(t : \boldsymbol{s}_\alpha) := \log_2 \frac{\mathbb{P}(\mathfrak{t})}{\mathbb{P}(\mathfrak{t}) - \mathbb{P}\left(\mathfrak{t} \cap \bigcap_{\boldsymbol{a} \in \alpha} \bar{\mathfrak{s}}_{\boldsymbol{a}}\right)} = \log_2 \frac{1}{p_{\boldsymbol{S}_\alpha|T}(s_\alpha \mid t)} .$$

**Proposition A.1.** *The misinformative part of the redundancy $I_\cap^{\text{sx}-}(T : \boldsymbol{S}_\alpha)$ vanishes if there exists a mapping $f : T \to \boldsymbol{S}$ from the target to the sources.*

*Proof.* If there exists a function $f : T \to \boldsymbol{S}$, all conditional probabilities of the form $p_{\boldsymbol{S}_\alpha|T}(s_\alpha \mid t) = p_{\boldsymbol{S}_\alpha|T}((s_{\boldsymbol{a}_{11}} \cap s_{\boldsymbol{a}_{12}} \cap \ldots) \cup \cdots \mid t)$ are equal to either one or zero. Thus, $I_\cap^{\text{sx}-}(T : \boldsymbol{S}_\alpha) = -\sum_{\boldsymbol{s},t} p_{\boldsymbol{S},T}(\boldsymbol{s},t) \log_2 p_{\boldsymbol{S}_\alpha|T}(s_\alpha \mid t) = 0$ for any $\alpha$. $\qquad \square$

To uniquely determine the label from a one-hot representation, it is sufficient to observe the one neuron that is equal to one. However, one gets the same information by observing all neurons which are equal to zero, since, by exclusion, the last one then has to be one. Let $\alpha_j := \{\{j\}\{1, ..., j-1, j+1, ..., n\}\}$ be the antichain describing the redundant information between the $j$-th source and the rest of the sources taken together. Further, let $I(Y : \vec{Y})$ describe the mutual information between a random variable $Y$ and its one-hot representation, which is trivially equal to its entropy $H(Y)$ due to the construction of $\vec{Y}$.

**Lemma A.3.** *The size of the local $I_\cap^{\text{sx}}$ redundancy $i_\cap^{\text{sx}}(y_j : \vec{y}_{\alpha_j})$ is $-\log_2 p_Y(y)$.*

*Proof.* Since the one-hot representation is a bijective function of the variable, Proposition A.1 implies that the misinformative part of the redundancy vanishes. The local $I_\cap^{sx}$ redundancy in question then amounts to $i_\cap^{sx}(y_j : \vec{y}_{\alpha_j}) = i_\cap^{sx+}(y_j : \vec{y}_{\alpha_j}) = -\log_2 p_Y(y_j \cup (y_1 \cap \cdots \cap y_{j-1} \cap y_{j+1} \cap \cdots \cap y_n)) = -\log_2 p_Y(y_j \cup y) = -\log_2 p_Y(y)$. $\qquad \square$

The atoms are ordered on a lattice by the partial ordering relation between atoms with antichain $\alpha$ and $\beta$ being given by (Williams & Beer, 2010)

$$\alpha \preceq \beta \Leftrightarrow \forall \boldsymbol{b} \in \beta \; \exists \boldsymbol{a} \in \alpha \text{ such that } \boldsymbol{a} \subseteq \boldsymbol{b}. \tag{17}$$

The atoms are, then, computed as the Moebius inversion of the corresponding redundancies on the lattice, which is referred to throughout the literature as the Redundancy Lattice.

**Lemma A.4.** *The degree of synergy $m \in \mathbb{N}$ increases monotonically on the redundancy lattice, i.e., $\alpha \preceq \beta \Rightarrow m(\alpha) \leq m(\beta)$.*

*Proof.* Let $\alpha, \beta \in \mathcal{P}(\mathcal{P}(\{1, \ldots, n\}))$ be two antichains on the $n$ redundancy lattice such that $\alpha \preceq \beta$. By definition of the degree of synergy (Equation (4)), there exists a set $\boldsymbol{b} \in \beta$ such that $m(\beta) = |\boldsymbol{b}|$. For this $\boldsymbol{b}$, it follows from the definition of the partial order of the antichains that there must then also exist a set $\boldsymbol{a} \in \alpha$ for which $\boldsymbol{a} \subseteq \boldsymbol{b}$. Thus, $\alpha \preceq \beta \Rightarrow m(\alpha) \leq |\boldsymbol{a}| \leq |\boldsymbol{b}| = m(\beta)$. $\qquad \square$

**Theorem A.4.** *The representational complexity of a categorical random variable $Y$ and its one-hot representation is equal to one.*

*Proof.* The local mutual information of the event $(T = t, S = s)$ amounts to $i(y : \vec{y}) = -\log_2(p_Y(y))$ and is thus equal to the local redundancy $i_\cap^{sx}(y_j : \vec{y}_{\alpha_j})$ (Lemma A.3). Because all local atoms are non-negative (Proposition A.1), all local atoms $\pi$ with antichains $\beta$ succeeding $\alpha_j$ must be zero. Since $m(\alpha_j) = 1$ and $m(\beta) = 1$ for all $\beta \preceq \alpha$ (Lemma A.4), the representational complexity of the one-hot encoding is $C = 1$. $\qquad \square$

## A.3 DNN implementation

### A.3.1 MNIST feedforward deep neural network

The MNIST networks analyzed in this paper are fully-connected feed-forward deep neural networks with quantized activation values, but float-precision weights. These networks are trained on the 60000 28x28 grayscale pictures of handwritten digits of the training set of the MNIST dataset (LeCun et al., 1998). To get better statistics for the test error as well as better estimates of information-theoretic quantities on the test set, the additional 50000 test samples of the QMNIST dataset have been added to the 10000 regular MNIST test set samples (Yadav & Bottou, 2019). The networks use tanh activation functions on the hidden layers, while on the output layer employing a softmax (for one-hot output layer) or sigmoid (for binary output layer).

Three different networks have been trained with 20 different random weight initializations each: Figures 3.B and 4.A have been computed on networks trained and evaluated with eight quantization levels per neuron but with different output layer representations: While the networks represented in Figure 3.B have a ten-neuron one-hot output layer, the networks whose results are depicted in Figure 4.A have only four output neurons of which each represents one bit of the binary representation of the numeric label. While the networks shown in Figures 5.B and D share the same network architecture as the ones in Figure 3, they have been trained and evaluated with only four quantization levels to make the computation of the coarse-grained PID more efficient.

Finally, the MNIST network with randomly assigned labels, as depicted in Figure 7 has the same network structure as the default MNIST network and has been trained for 5 different weight initializations with a different random shuffling of the train and test labels each. There we use eight quantization levels.

For all networks, we used stochastic gradient descent with a batch size of 64, learning rate of 0.01 and Xavier weight initialization. The networks with one-hot output representation employ a cross-entropy loss, while for the binary representation, a mean square error loss was chosen. Parameters and accuracies of the networks are summarized in Table 2.

| Setup | #runs | #quantization levels | Output rep. (#neurons) | train acc. | test acc. |
|---|---|---|---|---|---|
| Default | 20 | 8 | one-hot (10) | 99.96(2) | 95.0(4) |
| Binary Output | 20 | 8 | binary (4) | 99.59(4) | 95.4(2) |
| Reduced | 20 | 4 | one-hot (10) | 99.80(8) | 94.7(4) |
| Shuffled Labels | 5 | 8 | one-hot (10) | 30(3) | 10.2(2) |

Table 2: Network parameters and final accuracies for the three MNIST fully-connected DNNs referenced in this paper.

### A.3.2 CIFAR10 convolutional neural network

For the more complex CIFAR10 image classification task (Krizhevsky, 2009) we employed a larger feed-forward neural network with convolutional layers. The 50000 32x32x3 pictures are fed into three convolutional layers with 32, 64 and 128 filters, respectively and a kernel size of 3. The layers employ max pooling and ReLU activation functions. Afterwards, the result is flattened to 2048 numbers and fed into the fully-connected part of the network consisting of four layers with a width of 128, 32, 5, and 5, respectively, with tanh activation functions. These fully-connected layers are quantized using 8 quantization levels. Finally, the results are fed into the 10 neuron one-hot output layer.

Similar to the MNIST networks, the CIFAR10 network has been trained using stochastic gradient descent with a batch size of 64, a learning rate of 0.005, Xavier weight initialization and cross-entropy loss.

### A.3.3 Quantization schemes

In order to limit the information capacity of the networks, the activation values have been quantized to very few discrete values. For evaluation, the quantized activations $\ell$ are computed from the continuous values $\hat{\ell}$ as

$$\ell = \epsilon \left\lfloor \frac{\hat{\ell} - \sigma_{\min}}{\epsilon} \right\rceil + \sigma_{\min} \,, \tag{18}$$

where $\lfloor \cdot \rceil$ denotes rounding to the closest integer, the bin size $\epsilon$ is given by $\epsilon = (\sigma_{\max} - \sigma_{\min})/(n_{\mathrm{bins}} - 1)$ and $\sigma_{\min}$ and $\sigma_{\max}$ are the bounds of the activation function. This quantization scheme has been chosen as it reproduces the bounds exactly, i.e., $\hat{\ell} = \sigma_{\min/\max} \to \ell = \sigma_{\min/\max}$, and limits the rounding error to $\epsilon/2$.

For the training phase, the activation values of the forward pass are stochastically to make the training more robust. The stochastic scheme builds on the deterministic scheme presented before in that it rounds the values to the same value. However, whether values are rounded up or down to the nearest rounding point is no longer deterministic but given by a probability scaling linearly with the distance from the next two rounding points, giving

$$\ell = \epsilon\lambda + \sigma_{\min} \quad \text{where} \quad \lambda = \begin{cases} \left\lceil \frac{\hat{\ell}-\sigma_{\min}}{\epsilon} \right\rceil & r > \left( \frac{\hat{\ell}-\sigma_{\min}}{\epsilon} \mod 1 \right) \\ \left\lfloor \frac{\hat{\ell}-\sigma_{\min}}{\epsilon} \right\rfloor & r \leq \left( \frac{\hat{\ell}-\sigma_{\min}}{\epsilon} \mod 1 \right), \end{cases} \tag{19}$$

and $r \in [0, 1]$ is drawn i.i.d. from a uniform distribution for each neuron and each evaluation.

## A.4 Additional results

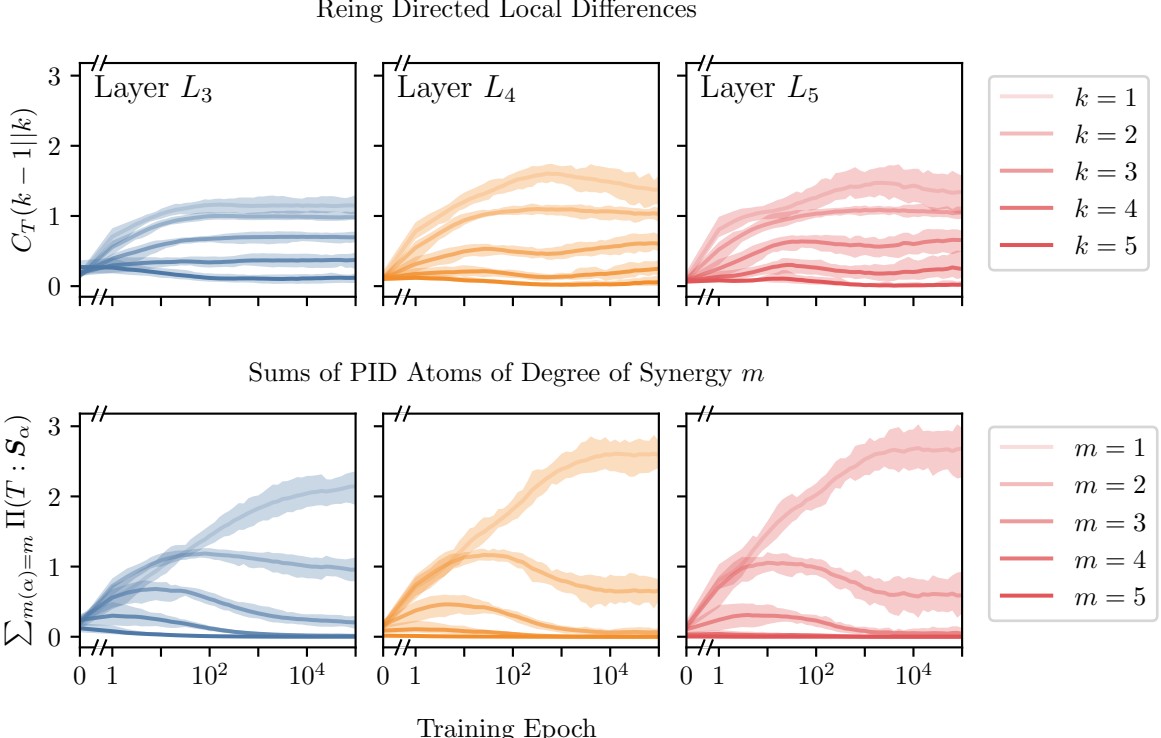

Figure 9: **Summing up PID atoms of a fixed degree of synergy shows more information is stored across single neurons than estimated by the directed local differences by Reing et al. (2021)**. The figure compares the sum of PID atoms of a certain degree of synergy (bottom row) with the directed local differences (top row) on three layers of the MNIST classifier network. Note that both decompositions for a given layer sum up to the total mutual information with the label.

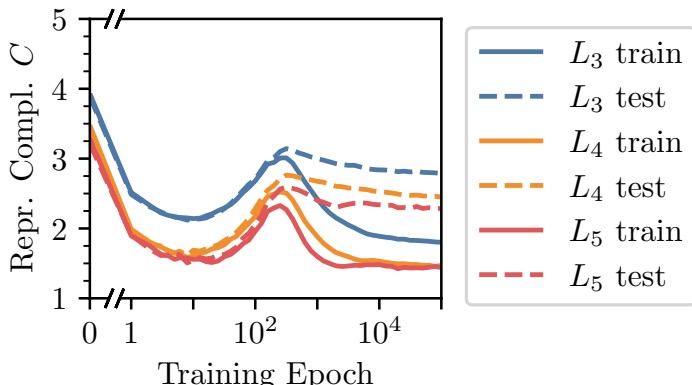

Figure 10: **Detailed comparison between the average representational complexity over 5 runs for the shuffled-label task described in Section 4.5 and Figure 7**. Shown are only the averages, but not the confidence intervals for better visibility. It is apparent that on this task that can only be solved by memorization of the random labels, the representation of the train dataset is much less synergistic than the one of the test dataset.

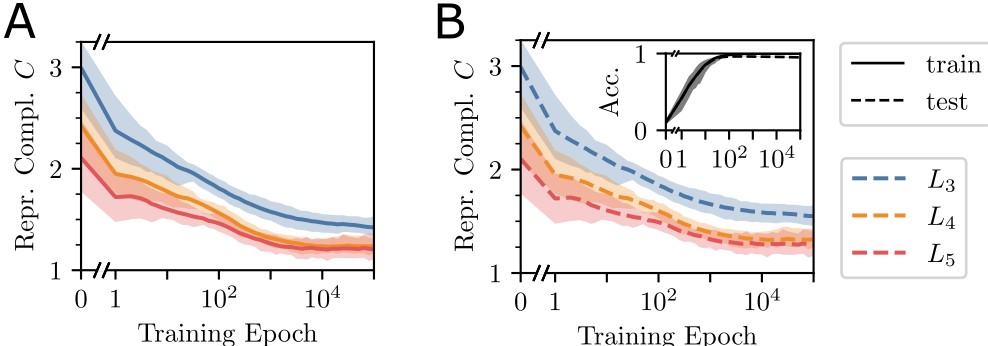

Figure 11: **Representational complexity computed on the train (A) and test (B) datasets follow the same general trend.** (A) is a copy of Figure 3B and was put here to allow easier comparison between the results on train and test datasets. The setup for this experiment is described in the caption of Figure 3, but also in Table 2. The comparison between train and test datasets can be found in the main text in Section 4.5.

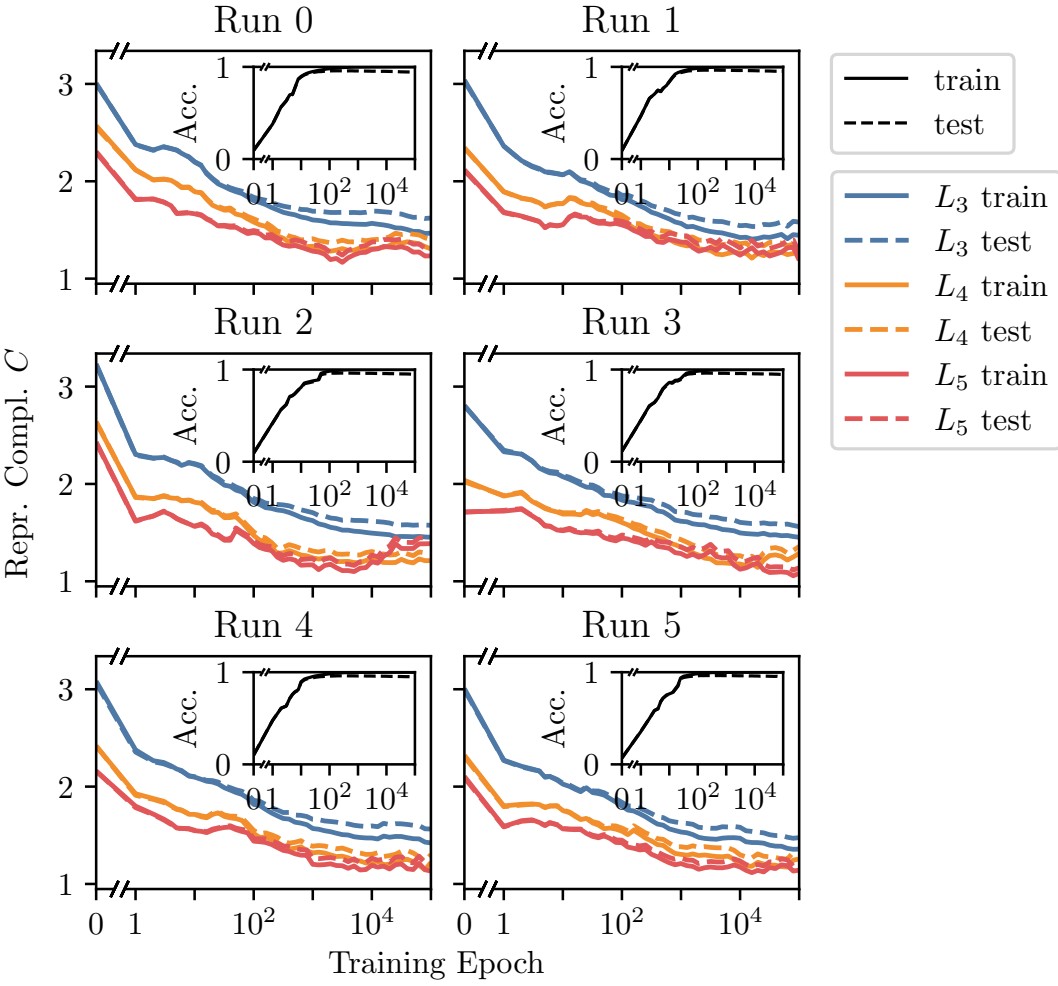

Figure 12: **Individual runs of the default MNIST setup show a consistent deviation between the representational complexity computed on the train and test datasets starting at around 100 epochs into training.** Shown are the first 6 out of overall 20 training runs with the same settings but different seeds.

