# OpenReview forum: "A Measure of the Complexity of Neural Representations based on Partial Information Decomposition"
_TMLR — Accepted by TMLR_

### Review · Reviewer_mLwm · 2023-02-04

**Summary Of Contributions:**

The paper proposes a new measure, called "representational complexity," which quantifies the number of intermediate-layer neurons one should inspect to make an inference about some target random variable. The paper then proposes several algorithms for approximating the represntational complexity when the number of neurons is large. Empirically, the paper reports that the quantity tend to reduce as the layer being inspected gets closer to the output, and as the training proceeds.

**Audience:**

No

**Broader Impact Concerns:**

I do not think there is any need to include additional broader impact statements.

**Claims And Evidence:**

No

**Requested Changes:**

- Please make it clear where the proposed quantity will be useful for. Validate the claim by comparing it to other baseline quantities.

- Please make the technical contributions clear in the main text.


**Strengths And Weaknesses:**

Strengths:
- The mathematical definition of the proposed representational complexity makes a lot of sense; I personally think this quantity has some potential to be a useful tool for the interpretation of the neural network architecture.

- The quality of writing is satisfactory, and the visual illustrations are very effective.

- The experiments are concerned with the "quantized" deep neural networks, without using any binning procedure; this is a good choice because binning obfuscates the message of the empirical result as it does not provide a good-enough approximation of the continuous quantities (as was discovered for information bottleneck principles).

Weaknesses:
- The paper does not clearly tell us where the proposed quantity is going to be useful for, and only gives some hints about it. The paper argues that understanding the decision of neural networks is important, but how exactly does this quantity help the purpose?

- Perhaps due to the aforementioned lack of clear use cases, the empirical results lack a baseline to compare to. Given some target application, how superior is the proposed measure when compared with other competing quantities, in terms of the interpretability? Why are the empirical results a "good sign" regarding the utility of the proposed quantity?

- Most of the formal theorem statements are hidden in the appendices, making the technical contribution of the paper very unclear.

---

> ### Author Response · Authors · 2023-02-22
> **About the usefulness of Representational Complexity**
>
> We thank the reviewer for his efforts and the valuable feedback, especially regarding the usecases of representational complexity. In the following we will respond to the weaknesses and requested changes.
>
> > The paper does not clearly tell us where the proposed quantity is going to be useful for, and only gives some hints about it. The paper argues that understanding the decision of neural networks is important, but how exactly does this quantity help the purpose?
>
> > Perhaps due to the aforementioned lack of clear use cases, the empirical results lack a baseline to compare to. Given some target application, how superior is the proposed measure when compared with other competing quantities, in terms of the interpretability? Why are the empirical results a "good sign" regarding the utility of the proposed quantity?`
>
> We understand the concerns of the reviewer and want to present some potential usecases, especially as a starting point for subsequent research. We are well aware of the narrow limits in terms of number of source variables when applying a full Partial Information Decomposition (PID) analysis to DNN representations, which might decrease its appeal drastically at first glance. However, we believe that very insightful research based on our work may contribute to better understand the internal representations of DNNs in the future.
>
> One of the clear benefits of our approach is that it allows for an assessment of the complexity of internal representations *with respect to a target variable*, in this case the label. This target gives our quantity important meaning in terms of the network's capability to represent the task-relevant label information. This is in contrast to many existing approaches, for example the local intrinsic dimension (LID), which does not focus on any specifically relevant aspect of the representation. When comparing both of these measures empirically - as we have done in this revision of the manuscript - we find that these two measures behave in similar ways, which is a "good sign" since both are used to answer a similar question. However in contrast to LID, the representational complexity decreases over subsequent layers in our experiments. Of course it is hard to draw conclusions at this point, but this consistent decrease matches the intuition that the layers in a DNN are optimized towards successively reducing the complexity of the task-relevant part of the representation. The relation to a target variable is important since just reaching a low-dimensional representation is not necessarily benefitial for solving the task. Harnessing this feature of being able to specificy a target random variable is a very promising avenue for better understanding neural representations in the future.
>
> Second, we believe representational complexity could also be used to study the phenomenon of overfitting. We compared the representational complexity of the train and test datasets (section 4.5) and find that they start to differ consistently towards later stages of training. This reveals a difference in the representation of samples that the network was trained on compared to previously unseen ones. We further investigate this phenomenon in the extreme case of training on randomly assigned labels and find that after a temporary increase in representational complexity around the time of the onset of performance, the complexity for the samples of the test dataset is significantly higher. We are hesitant to draw any conclusions without further research, but are convinced that this behaviour might be relevant for a better understanding of overfitting.
>
> Third, in our research we find that the average degree of synergy of the encoding of the label information is changing substantially over the course of training. This allows for an assessment whether and when a lower-order analysis might be able to capture most of the mutual information with the label, and when it would miss much of the information encoded in a higher-order manner. We find that for the networks we look at, approaches based on analyzing information in pairs of neurons would fail to capture a large part of the relevant information during the early stages of training.
>
> Last, to our knowledge it has previously not been achieved to apply a full PID analysis to up to five source variables, and one reason for that may well be the extremely fast growing number of individual information atoms that need to be considered. With the sheer vastness of individual contributions that are intrinsic to the PID problem, extracting a useful interpretation for more than two or three sources becomes a challenging task. In our work we show up a way to combine the expressivity of the PID framework with an intuitive interpretation by combining the atoms in a meaningful way. We expect that subsequent work will build on this approach and construct other meaningful quantities in a similar manner.

---

> > ### Author Response · Authors · 2023-02-22
> > **Clarifying technical contributions and providing additional comparisons**
> >
> > > Most of the formal theorem statements are hidden in the appendices, making the technical contribution of the paper very unclear.
> >
> > We adress this criticism below.
> >
> > > Requested Changes:
> > Please make it clear where the proposed quantity will be useful for. Validate the claim by comparing it to other baseline quantities.
> >
> > We incorporated more of the arguments mentioned above about the usefulness of representational complexity into the discussion in the manuscript.
> >
> > We also added empirical comparisons with two other quantities. On the one hand we chose the local intrinsic dimension (LID) (section 4.4), a non-information-based quantity that has been used in the literature to analyze representations in DNNs and tries to answer a related question: How many dimensions are needed locally to describe most of the covariance of the representation? We argue that the possibility to specify a target variable is one of the most important features of representational complexity that existing approaches like LID lack, leading to less meaningful and interpretable results.
> >
> > On the other hand, we compare the representational complexity to Neural Information Decomposition (NID) (section 4.3), that also specifies a target random variable. NID is a very similar information-theoretic quantity that can like ours be used to estimate the amount of mutual information with the label that is encoded in higher-order relations between the sources. However, it does trade in some of the expressiveness of the PID framework for easier practical application. We believe that both approaches are valid, however find that NID tends to overestimate the amount of information in higher-order relations.
> >
> > >Please make the technical contributions clear in the main text.
> >
> > We agree that the way our technical contributions were presented in the previous version of the manuscript, it was not clear enough to the reader. We tried to adress this point by adequate changes to the manuscript, by adding more of the technical contributions to the main text. We added more explanation in the beginning of the appendix and explain which parts are solely for the purpose of giving more theoretical background, which parts are novel proofs and which parts are details on the experiments and additional results. We additionally want to point to the last paragraph of the introduction, were we specifically explain the contributions of our work.
> >
> > ----
> >
> > Overall, we again thank the reviewer for his efforts and hope that the changes we made allow the reader to better assess the contributions that our work offers to the community. We also hope that the evidence presented for our claims is more convincing after adding comparisons with two related quantities.

---

### Review · Reviewer_MWVV · 2023-02-08

**Summary Of Contributions:**

The authors leverage a decomposition of Mutual Information quantities (called Partial Information Decomposition) to define an interpretable measure of the complexity of Neural representations. This measure is called "Representational Complexity" and is computed in some toy examples as well as empirically in some deep neural network models.

**Audience:**

Yes

**Broader Impact Concerns:**

I do not see any particular ethical implication for this work.

**Claims And Evidence:**

No

**Requested Changes:**

I would suggest conforming to standard notation and use $I(T\textbf{;}S)$ rather than $I(T\textbf{:}S)$.

I believe that the section on PID should be expanded as now is quite short and not sufficiently self-contained, perhaps moving a more thorough exposition to the appendix. The space gained should be employed to expand upon the theoretical analysis of the quantities introduced with a formal treatise of examples. Some examples in which the random variables are continuous should be included. The behavior of $C$ in extreme settings should be explicitly written (as the ones proposed in the "Strengths and Weaknesses" part of the review).

More comparisons should be advanced in order to justify the choice of (5) rather than any other approach which is already present in the literature.

Since the code cannot be shared (although I don't see why one cannot make a novel anonymous GitHub profile just to share the code for the purpose of the review) more details on the implementations for Section 4 should be included.

The paper should either be presented as a theoretical treatise of some new quantity or as experimental. Wherever the choice falls, relative sections must be expanded. So far it is lacking both in theory and in experiments. Two simple examples are not enough without theory and if more cannot be provided (due to computational constraints, which is an important weakness of the approach, strongly limiting its applicability in concrete settings) then the theory should be stronger.

Often unclear terminology is used to push formal arguments. I would suggest either removing said comments or bringing forward more formal statements. One example is in Section 4.2.Subsampling: the authors claim that subsampling suffers from conceptual flaws but I cannot follow the argument as the terminology used can mean multiple things and does not possess formally agreed-upon definitions (lack of rigor). Similarly, Appendix A.1 is very confusing. The first half of the "proof" is unclear: monotonicity is part of the definition of parthood distribution (Definition 1.(iii) of Gutknecht et al, https://arxiv.org/pdf/2008.09535.pdf) why would it need to be proven, in what looks like a tautological argument? The second half needs to be expanded with more formal arguments. Analogously, Appendix A.2 faces similar issues.
To understand the quantity in (3) the reader is referred to Appendix A.2 making it look like the topic is self-contained but it actually isn't. I would suggest expanding the Section.

Whenever a reference is given for a formal argument, please refer to the specific result (rather than the entire paper, which can be a significant amount of pages long). For example, shortly before equation (7) Gutknecht et al is referred but it is not easy to find the specific argument. Similarly for Equation (8), which should be justified through Appendix A.1 and (Williams and Beer, 2010): where in that work?

The subsection on Coarse-graining (proposed as an alternative to Subsampling) also needs to be meaningfully expanded and justified. Especially since it is being used to provide experimental settings to justify the proposal of (5) as a reasonable quantity.

I strongly suggest removing all the sentences in which MI is endowed with a meaning that does not belong to it: e.g., Section 3.2: "Since the Mutual Information term $I(T:S_i)$ captures all information that the single source $S_i$ carries about $T$ ..." what does 'capturing information' mean?


**Strengths And Weaknesses:**

The idea of manipulating Mutual Information in order to overcome its inherent limitations when applied to neural networks is interesting. However, despite the title, it is hard to find "rigor" in this paper or a conclusive statement of any sort (beyond the plots).

The paper seems based on two assumptions: Mutual Information (MI) measures information and it is limited (as measuring information, it should satisfy a set of properties that are enforced upon it via this decomposition). This is simply $\textbf{not}$ true: Mutual Information appeared as a solution to a very practical problem in communication. As a convex functional of measures, it seems to satisfy a series of properties that are useful in a variety of settings (properties that follow from the Kullbacak-Leibler divergence of which the Mutual Information is a specific instance). It has never been claimed anywhere that Mutual Information should measure information (even because the matter is more philosophical than mathematical at this point, what is even ``information''?). One is thus not measuring information in the layers but rather how much they depend on the input (seeing MI as a measure of $\textit{dependence}$, following the axiomatic approach introduced for instance by Rényi in his work published in 09/1959 and titled "On measures of dependence"). Also, later on in the paper (Appendix A.1), the word "predictability" is associated with Mutual Information which is also false in general. The property being used is essentially just "Data-Processing Inequality" which is a consequence of convexity and the amount of stochasticity induced when applying Markov Kernels (intuitively, if I apply more randomness e.g., adding noise, then the variables will depend less on each and Mutual Information decreases). This is true of $\textbf{any}$ divergence measure which is a convex functional. In order to make such a claim one would have to introduce a notion of "predictability" and provide results that show how the MI is involved in such an operational problem. This assumption seems relevant in multiple areas of the paper and in particular Section 3.2. "We propose that this difficulty of retrieving information may be quantified by considering the minimum number of sources that are needed jointly in order to reveal a piece of information". This is very informal, attaches meaning to MI, and seems much more related to deterministic mappings (e.g., how many "neurons" seen as mappings, do I need to recover the input exactly?) than to MI, which is a quantity measuring dependence in stochastic settings and, by extension, also to deterministic mappings in $\textbf{discrete}$ settings.

I find most of the contributions under-explored. The Representational Complexity is introduced in (5) but not properly defined. It is unclear what type of object it is, and what properties it satisfies. Is it an information measure itself? What properties does it satisfy?
In the "Example applications of representational complexity" the intuition is nice, but there is a lack of formality and rigor. What are the details of the computations of $C$ for the various setting? How can a reader (or in this case, a reviewer) check them?
It also seems that most of the limitations of using MI still persist. What happens to $C$ when $T$ is a deterministic function of one of the $S$'s? Does the quantity blow up? Is it not defined? What are the measurability requirements necessary to define $C$? When is it well defined? When is it not?

A comparison section is missing: why is this Representation Complexity better than other approaches? Why is Rosas' paper mentioned in the Limitations but not in related work? Why is there no direct comparison with the decomposition provided in that work (which, at least at a glance, looks rigorous)?

In order to provide a rigorous and theoretically sound decomposition one should at least formally compare with (a few) others and argue why their approach is better/more fitting. Why have other options not been explored? Otherwise, it feels arbitrary. A few examples don't make for a theoretical argument but they can, at best, back a theory up. If the approach is presented as something meant to be used in practice (that would be perfectly fine as long as it is thoroughly justified with toy settings and real-world settings) but then it cannot be employed because of big computational issues then the theory should be much stronger in order to push the community to address said limitations. Otherwise, I see no real contribution in introducing a practical approach that cannot be used in practice.

---

> ### Author Response · Authors · 2023-02-22
> **Response to issue of "rigorousness"**
>
> We would like to thank the reviewer for the thorough and thoughtful revision of the paper. Though the revision feels overly harsh at times, we appreciate it and highly value the effort and time that the reviewer spent on this manuscript. In the following, we will address each point raised. For the points where we think the reviewer has a misunderstanding, we not only respond and clarify them but have also made changes accordingly to the main text to make these points more clear to future readers. For the other points we will detail the changes made in the main text to address them. We think that both types of critisim were constructive and helped improving the manuscript. Thus we again thank the reviewer for their efforts.
>
> > The idea of manipulating Mutual Information in order to overcome its inherent limitations when applied to neural networks is interesting. However, despite the title, it is hard to find "rigor" in this paper or a conclusive statement of any sort (beyond the plots).
>
> The term "rigorous" used in the original manuscript title was meant to refer to the clear interpretability of our measure, bestowed on it by being built on the very expressive framework of partial information decomposition. Nevertheless, we understand that the term "rigorous" might induce expectations of a more thorough mathematical discussion of the quantity. For this reason, we have removed the claim from the title, while at the same time extending the discussion of the mathematical properties of representational complexity in the manuscript.

---

> > ### Author Response · Authors · 2023-02-22
> > **What is meant by "information"**
> >
> > > The paper seems based on two assumptions: Mutual Information (MI) measures information and it is limited (as measuring information, it should satisfy a set of properties that are enforced upon it via this decomposition). This is simply not true: Mutual Information appeared as a solution to a very practical problem in communication. As a convex functional of measures, it seems to satisfy a series of properties that are useful in a variety of settings (properties that follow from the Kullbacak-Leibler divergence of which the Mutual Information is a specific instance). It has never been claimed anywhere that Mutual Information should measure information (even because the matter is more philosophical than mathematical at this point, what is even ``information''?). One is thus not measuring information in the layers but rather how much they depend on the input (seeing MI as a measure of dependence, following the axiomatic approach introduced for instance by Rényi in his work published in 09/1959 and titled "On measures of dependence").
> >
> > The premise of the manuscript is to view DNNs in terms of their information processing capabilities. Throughout the manuscript, the term "information" is meant to be understood in the narrow sense of Shannon information. We added a brief section to the manuscript to differentiate this notion of information from other concepts of (e.g. semantic) information and clarify how the term is used in our paper.
> >
> > In what follows we will lay out how Shannon information, and in particular Partial Information Decomposition (PID), can be used to gain insight into the internal representations formed by the activations of Deep Feedforward Neural Networks (DNNs) performing a supervised classification task. The first layer of the DNN is fed samples from a finite training dataset. Each subsequent layer receives the input from the previous layer (input), performs a non-linear transformation and passes the result (in our case a discretized version of it) on to the next layer. The network is then optimized for producing an output that best matches the corresponding label. Note that each individual layer can be viewed as a Shannon information channel, and as pointed out by the reviewer, obeys the data-processing inequality.
> >
> > In all supervised datasets used for the experiments in this manuscript, there exists a surjective mapping from input samples to labels, thus the mutual information between input and label is equal to the entropy of the label. The goal of the network optimization is to approximate this mapping defined by the dataset. When the inputs are processed by the network, each subsequent layer will have less mutual information with the label (in other words: lose task-relevant information), and the remaining mutual information of the last layer with the label gives an upper bound on the prediction accuracy of the network via Fano's inequality (see Geiger et al. 2021, Appendix IV). The success of optimization is thus dependent on increasing this mutual information. Furthermore, the widespread use of a cross-entropy loss function in DNNs, which itself is a concept related to Shannon information theory, further motivates that analyzing Shannon information in earlier layers might reveal interesting facts about the learning dynamics. Using information theory for these information channels in the context of the label, and in particular mutual information is thus sound.
> >
> > Using the relatively novel framework of PID it is possible to go beyond the channel: decomposing the total mutual information of a (small enough) layer with the label into unique, redundant and synergistic contributions of individual neurons, thus giving insight into how the mutual information is represented among the neurons. One of the main contributions of this manuscript is to combine the numerous atoms in a way that allows for a meaningful interpretation as a specific notion of complexity via the average degree of synergy.

---

> > > ### Author Response · Authors · 2023-02-22
> > > **Resolving the concern about predictability**
> > >
> > > > Also, later on in the paper (Appendix A.1), the word "predictability" is associated with Mutual Information which is also false in general. The property being used is essentially just "Data-Processing Inequality" which is a consequence of convexity and the amount of stochasticity induced when applying Markov Kernels (intuitively, if I apply more randomness e.g., adding noise, then the variables will depend less on each and Mutual Information decreases). This is true of any divergence measure which is a convex functional. In order to make such a claim one would have to introduce a notion of "predictability" and provide results that show how the MI is involved in such an operational problem.
> > >
> > > We are unsure whether the reviewer thinks that predictibaility is not related to MI at all or whether their issue is that the term "predictability" was used without sufficient explanation.
> > >
> > > In the following, we are going to explain how MI is related to predictability and then comment on the additional insights that this adds to the mathematical interpretation. In order to understand the reation of MI and predictability, we need to look at the pointwise mutual information, i.e. $\log \frac{p(t\mid s)}{p(t)}$. Following Robert Fano's argumentation, this pointwise mutual information quantifies how much a recieved symbol $s$, i.e. the realization of the event $\{S=s\}$ is informative about the true sent symbol $t$, i.e. the realization of the event $\{T=t\}$, via the channel. Thus if the probabilty of guessing that the sent symbol is $t$ given recieving the symbol $s$, i.e. $p(t\mid s)$, is larger than the prior probability of guessing $t$, i.e. $p(t)$ then recieving $\{S=s\}$ is informative about the actual sent $\{T=t\}$ and in this case the pointwise mutual information is positive. However, when the pointwise mutual information is negative, the probability of guessing that the sent symbol is $t$ given recieving the symbol $s$, i.e. $p(t\mid s)$, is smaller than the prior probability of guessing $t$, i.e. $p(t)$ then recieving $\{S=s\}$ is misinformative about the actual sent $\{T=t\}$ as its deminished the posterior belief that $t$ was sent. Note that some argue that the term misinformative can be misleading and thus a differennt term needs to be used to describe it, however, the concesus is that pointwise mutual information can be interpreted in terms of inference. This carries on to the average MI and hence our usage the term predictibility. Similarly to the DPI where the term predictibilty was used, the convexity explains mathematically why the DPI holds however an interpretation is required and here one could see it as looking at the full recieved codeword will improve the prediction of the sent symbol as opposed to looking at only parts of it. This also goes in hand with arguments from coding theory in praticuluar erasure channels/codes.
> > >
> > > We see that this predictibility in the appendix might have been misleading and didn't serve the purpose of giving a further intuition to the reader about DPI beyond the mathematical reasoning. Since we think it is a minor point we decided to rephrase it, despite its soundness.
> > >
> > > > This assumption seems relevant in multiple areas of the paper and in particular Section 3.2. “We propose that this difficulty of retrieving information may be quantified by considering the minimum number of sources that are needed jointly in order to reveal a piece of information”. This is very informal, attaches meaning to MI, and seems much more related to deterministic mappings (e.g., how many “neurons” seen as mappings, do I need to recover the input exactly?) than to MI, which is a quantity measuring dependence in stochastic settings and, by extension, also to deterministic mappings in discrete settings.
> > >
> > > The piece of information here is the part of correlation that neuron extracts from what it recieves about the ground-truth label. It never reveals information about the input. The aboutness here is always the task-relevent information that is the label in the case of supervised learning.
> > >
> > > In the last point, the reviewer seems to suggest that we analyze the mutual information of the hidden layers with the input ("...recover the input..."). This is incorrect: In our manuscript, we decompose the mutual information between the hidden layers and the ground-truth classification label, ensuring that we only consider task-relevant correlations and ignore the irrelevant additional variations of the input variable. Furthermore, note that representational complexity reflects the *average* number of neurons needed to retrieve a certain piece of information - if one wants to recover *all* of the information ("...recover the input *exactly*"), more (and often all) source variables are needed.

---

> > > > ### Author Response · Authors · 2023-02-22
> > > > **Well-definedness and the properties of Representational Complexity**
> > > >
> > > > > I find most of the contributions under-explored. The Representational Complexity is introduced in (5) but not properly defined. It is unclear what type of object it is, and what properties it satisfies.
> > > >
> > > > Despite disagreeing with the claim of underexploration, we think that the queries made by the reviewer are valuable. We added a paragraph discussing the definition of RC and its mathematical properties.
> > > > > Is it an information measure itself?
> > > >
> > > > RC is not an information measure. It is the expected degree of synergy over all information atoms that decompose the mutual information with the label. As it is normalized by the total mutual information, it does not constitute a measure for information itself but is proposed as a complementary quantity that captures *how* the mutual information is spread amongst multiple source variables. As such, it constitutes a unitless interpretable summary statistic computed from the PID atoms.
> > > > > What properties does it satisfy?
> > > >
> > > > Since $C$ is an information theoretic average, it inherits various properties that PID and MI exhibit. Some of these evident properties are: invariance under isomorphisim and symmetry as invariance of reordering of sources. Other properties depend on the PID measure that is used. For the one used in this paper, one can easily see that continuity and differentiability as a function of the joint probability distribution $p(t,s)$ immediatly follow. Finally, there are the upper and lower bounds for representational complexity which are 1 and the maximum number of neurons in the layer.
> > > >
> > > > > In the "Example applications of representational complexity" the intuition is nice, but there is a lack of formality and rigor. What are the details of the computations of C for the various setting? How can a reader (or in this case, a reviewer) check them?
> > > >
> > > > The code to reproduce the main empirical results can be found in the supplementary material to the original TMLR submission. A detailed explanation of how to compute SxPID atoms by hand is beyond the scope of this paper, and we refer the reviewer to the SxPID reference implementation by the original author (which is included in the supplementary materials or alternatively available as part of the IDTxl package) and our functions for computing the representational complexity from these atoms. In the particular case of the one-hot encoding where a computation is infeasible, the value for the representational complexity is derived mathematically in the appendix of our manuscript.
> > > >
> > > > > It also seems that most of the limitations of using MI still persist. What happens to C when T is a deterministic function of one of the S's? Does the quantity blow up? Is it not defined?
> > > >
> > > > If $T$ is a deterministic function of one of the $S$'s then MI is $H(T)$ so the mathemtically the quantity has no undefindness problems in the discrete case. Now when looking at the PID terms since $T$ is a function of one of the S's then the conceputaully one expects that only PID atoms that could be nonzero are either redundancy with that $S_i$ and the unique information of that $S_i$, resulting in a representational complexity of $C=1$. So to answer your question in this case $C$ is well defined because the PID terms and the MI are well defined.
> > > >
> > > > > What are the measurability requirements necessary to define C? When is it well defined? When is it not?
> > > >
> > > > There is an implicit requirement of $C$ which is that $MI(T: S) \neq 0$. This is because when $MI(T: S)=0$ the representational complexity is intuitvely undefined since the $S$ do not represent $T$ at all. Now regarding  the measurability requirements. Assuming that $MI(T:S)\neq 0$, in the discrete case, $C$ needs only a single measurable requirement which is $T$ and $S$ are random variables. In the continuous case one needs to additionally consider the case when $MI(T: S)$ is infinite. As most PID measures do not yield interpretable results in this case, the RC computed from these is similarly vacuous.  To conclude, in the discrete case $C$ is well defined and has loose measurable requirments whereas in the continuous case $C$ is strictly defined when $MI(T:S)< \infty$. We added a small paragraph discussing the well-defindness of $C$ in our manuscript.

---

> > > > > ### Author Response · Authors · 2023-02-22
> > > > > **Addressing comparisons to other measures**
> > > > >
> > > > > > A comparison section is missing: why is this Representation Complexity better than other approaches?
> > > > >
> > > > > We have added a new comparison section and compared $C$ with two other measures. While we cannot claim "superiority" over other approaches, we show that our approach based on PID gives slightly different results than a related information-theoretic approach by Reing et al. which only relies on classical information theory. We have furthermore added a comparison to an approach based on local dimensionality and compare $C$ to this measures on the basis of complemenatarity. We have detailed in the manuscripts the points for which $C$ has an added value compared to the other measures.
> > > > >
> > > > > > Why is Rosas' paper mentioned in the Limitations but not in related work? Why is there no direct comparison with the decomposition provided in that work (which, at least at a glance, looks rigorous)?
> > > > >
> > > > > The work of Rosas presents a synergy based measure that is operationally well motivtated. However, the measure does not statisfy the so-called *consistency equations*, meaning that the mutual information quantities with subsets of the sources cannot be constructed from the atoms, which is an important criterion for a PID measure. So until a modified version of this measure is developed, we cannot use it as a PID measure since the resulting atoms are not necessiraly the comparable to a consistent PID definition. Secondly in that paper Rosas el al. define the backbone atoms which are similar to the grouping that we make in $C$ in order to compute the average. However, our gouping comes from atoms computed using a redundancy-base measure and are thus distinct from Rosas' backbone atoms in their interpretation and in the way they are computed. Nevertheless, while we pointed out that such grouping is related to Rosas' in the **conclusion and outlook** section in the original manuscript, we have made the comparison more prominent in the updated version.
> > > > > Nevertheless, we agree with the author that an empirical comparison with Rosas' measure can provide added value to the paper. Unfortunately, we were unable to complete such an analysis in the brevity of time granted to us for this reply. If, however, the reviewer signals that this comparison is essential for the chance of success of this paper, we are prepared to produce such an additional comparison within short notice (2 weeks).

---

> > > > > > ### Author Response · Authors · 2023-02-22
> > > > > > **Addressing requested changes: Notation, behaviour of C, comparisons and Code**
> > > > > >
> > > > > > > I would suggest conforming to standard notation and use $I(T;S)$ rather than $I(T:S)$
> > > > > >
> > > > > > The colon notation is a convention widely adopted in the PID community. While this convention deviates from the notation used in classical information theory textbooks, we do not expect this choice to cause confusion for potential readers and thus see no immediate necessity to change it.
> > > > > >
> > > > > > > I believe that the section on PID should be expanded as now is quite short and not sufficiently self-contained, perhaps moving a more thorough exposition to the appendix. The space gained should be employed to expand upon the theoretical analysis of the quantities introduced with a formal treatise of examples.
> > > > > >
> > > > > > To make the concepts of PID more accessible to the audience, we have rephrased and slightly expanded upon the corresponding chapters in the appendix.
> > > > > >
> > > > > > > Some examples in which the random variables are continuous should be included.
> > > > > >
> > > > > > The existing PID measures for continuous variables are very limited and mostly underdeveloped. The main devoloped continuous measure is the $I_{\mathrm{broja}}$, however, it is unfortuently only for two source variables. We prefer to only start including continuous measures once the PID field is mature enough to handle continuous variables, which is currently not the case.
> > > > > >
> > > > > > > The behavior of $C$ in extreme settings should be explicitly written (as the ones proposed in the "Strengths and Weaknesses" part of the review).
> > > > > >
> > > > > > Unfortunately, it has been unclear to us which "extreme" settings other than the ones already discussed in the toy examples section of the manuscript the reviewer would like to see discussed.
> > > > > >
> > > > > > > More comparisons should be advanced in order to justify the choice of (5) rather than any other approach which is already present in the literature.
> > > > > >
> > > > > > We agree with this criticism and have added two measures to compare our results to. please refer to our reply in the Strengths and weaknesses
> > > > > >
> > > > > > > Since the code cannot be shared (although I don't see why one cannot make a novel anonymous GitHub profile just to share the code for the purpose of the review) more details on the implementations for Section 4 should be included.
> > > > > >
> > > > > > We included the nninfo python package with scripts to reproduce the main empirical findings of our paper in the supplementary material to the original submission. However, we apologize for overlooking to mention the supplementary material in the original manuscript. We additionally see how instead referring to our future plans to make the nninfo package a public tool for information-theoretic analysis in DNNs might have given the wrong impression that we have not shared our code with the reviewers.
> > > > > >
> > > > > > > The paper should either be presented as a theoretical treatise of some new quantity or as experimental. Wherever the choice falls, relative sections must be expanded. So far it is lacking both in theory and in experiments. Two simple examples are not enough without theory and if more cannot be provided (due to computational constraints, which is an important weakness of the approach, strongly limiting its applicability in concrete settings) then the theory should be stronger.
> > > > > >
> > > > > > We hope the reviewer looks at the changes that were made to strengthen both the theortical and practical ascpects.

---

> > > > > > > ### Author Response · Authors · 2023-02-22
> > > > > > > **Addressing unclear terminology and unclear parts of the manuscripts**
> > > > > > >
> > > > > > > > Often unclear terminology is used to push formal arguments. I would suggest either removing said comments or bringing forward more formal statements One example is in Section 4.2.Subsampling: the authors claim that subsampling suffers from conceptual flaws but I cannot follow the argument as the terminology used can mean multiple things and does not possess formally agreed-upon definitions (lack of rigor)
> > > > > > >
> > > > > > > We rewrote this paragraph and hope it is more clear.
> > > > > > >
> > > > > > > > Similarly, Appendix A.1 is very confusing.
> > > > > > >
> > > > > > > We made some changes to this paragraph to make less confusing.
> > > > > > >
> > > > > > > > The first half of the "proof" is unclear: monotonicity is part of the definition of parthood distribution (Definition 1.(iii) of Gutknecht et al, https://arxiv.org/pdf/2008.09535.pdf) why would it need to be proven, in what looks like a tautological argument? The second half needs to be expanded with more formal arguments.
> > > > > > >
> > > > > > > We reworked the appendix to clarify which sections are meant to provide additional theoretical background for PID and which constitute novel proofs. Assuming the reviewer is referring to Appendix A.1 of the original manuscript, this section was meant to provide a brief and intuitive introduction to the work of Gutknecht et al. to clarify the different ways of indexing PID atoms to the reader. We appreciate the reviewers feedback and have rewritten this section.
> > > > > > >
> > > > > > > > Analogously, Appendix A.2 faces similar issues. To understand the quantity in (3) the reader is referred to Appendix A.2 making it look like the topic is self-contained but it actually isn't. I would suggest expanding the Section.
> > > > > > >
> > > > > > > We are unsure what exactly the reader finds unclear in Appendix A.2. However, similar to the previous appendix section, we made some changes to make intention of the section and the language more clear.
> > > > > > >
> > > > > > > > Whenever a reference is given for a formal argument, please refer to the specific result (rather than the entire paper, which can be a significant amount of pages long). For example, shortly before equation (7) Gutknecht et al is referred but it is not easy to find the specific argument. Similarly for Equation (8), which should be justified through Appendix A.1 and (Williams and Beer, 2010): where in that work?
> > > > > > >
> > > > > > > We thank the reviewer for point this out. Neccessary changes have been made to fix this issue.
> > > > > > >
> > > > > > > > The subsection on Coarse-graining (proposed as an alternative to Subsampling) also needs to be meaningfully expanded and justified. Especially since it is being used to provide experimental settings to justify the proposal of (5) as a reasonable quantity.
> > > > > > >
> > > > > > > We would be grateful if the reviewer could elaborate on what they feel is missing from this section.
> > > > > > >
> > > > > > > > I strongly suggest removing all the sentences in which MI is endowed with a meaning that does not belong to it: e.g., Section 3.2: "Since the Mutual Information term $I(T:S_i)$ captures all information that the single source $S_i$ carries about $T$..." what does 'capturing information' mean?
> > > > > > >
> > > > > > > We have added a paragraph to avoid any confusion that could arise of what exactly mean by information. We refer the reviewer to the response to a similar point in the **strengths and weaknesses** on the reasonability of using the term information. We hope that this is satisfactory to the reviewer.
> > > > > > >
> > > > > > > We thank the reviewer again for his thorough analysis and feedback to our manuscript.

---

> > > > > > > ### Comment · Reviewer_MWVV · 2023-03-02
> > > > > > > **Comparisons**
> > > > > > >
> > > > > > > I appreciate that the authors provided some comparisons with other approaches,
> > > > > > > however, I do not fully grasp the conclusions of said comparison and I do not believe
> > > > > > > they should be left to the reader's speculation only.
> > > > > > > "While this Reing representational complexity does decrease in the earlier chapters of training, it stays higher than the
> > > > > > > PID based representational complexity, shows more fluctuations and does not decrease throughout successive
> > > > > > > hidden layers (Figure 6A)"
> > > > > > > It stays higher, but it is not clear to me what the minimal value it should achieve is.
> > > > > > > The fact that it is higher in absolute terms should not be a flaw unless there is evidence
> > > > > > > that a quantity measuring what the authors are trying to measure (and satisfying a set of
> > > > > > > assumed properties) should be below a certain value.
> > > > > > > As for the lack of ordering throughout successive hidden layers,
> > > > > > > a natural question is: can you guarantee a consistent decrease for your measure?
> > > > > > > The measure $C_\text{Reing}$ feels quite similar to the one you proposed, except for that lack
> > > > > > > of ordering in the first epochs. Couldn't there be a setting where your measure does the same?
> > > > > > > I think the phenomenon should be explored further.

---

> > > > > > > > ### Author Response · Authors · 2023-03-08
> > > > > > > > **Comparisons**
> > > > > > > >
> > > > > > > > > I appreciate that the authors provided some comparisons with other approaches, however, I do not fully grasp the conclusions of said comparison and I do not believe they should be left to the reader's speculation only. "While this Reing representational complexity does decrease in the earlier chapters of training, it stays higher than the PID based representational complexity, shows more fluctuations and does not decrease throughout successive hidden layers (Figure 6A)" It stays higher, but it is not clear to me what the minimal value it should achieve is. The fact that it is higher in absolute terms should not be a flaw unless there is evidence that a quantity measuring what the authors are trying to measure (and satisfying a set of assumed properties) should be below a certain value.
> > > > > > > >
> > > > > > > >
> > > > > > > > We agree with the reviewer that stating that the $C_{\mathrm{Reing}}$ measure stays higher might easily be misinterpreted as claiming a shortcoming of Reing's procedure, while our intention was only to point out that there exists an empirical difference to our measure. We observe that $C_{\mathrm{Reing}}$ does not decrease consistently over the course of training. This first and foremost shows that $C_{\mathrm{Reing}}$ and our Representational Complexity disagree not only in their precise value but also in their qualitative behaviour. We argue the interpretation of our measure of Representational Complexity to be more strongly supported, as it is directly based on the clear interpretations of the PID atoms and allows for a clear separation of different degrees of synergy.
> > > > > > > >
> > > > > > > > > As for the lack of ordering throughout successive hidden layers, a natural question is: can you guarantee a consistent decrease for your measure? The measure $C_\text{Reing}$ feels quite similar to the one you proposed, except for that lack of ordering in the first epochs. Couldn't there be a setting where your measure does the same? I think the phenomenon should be explored further.
> > > > > > > >
> > > > > > > > Please note that the ordering of $C$ throughout successive hidden layers is *not* a property of the measure but rather a property that we empirically observe for DNNs on average. Our intuitive explanation for this phenomenon is that the decrease over successive layers might be reflective of the network representing the relevant information in a more and more simple way. For a counterexample where $C$ does not decrease, consider the example in Figure 2A and B: Since the same amount of information is encoded in Figure 2A and 2B, one could imagine a hand-constructed network in which the first layer has a one-hot representation (2A) and the second layer has a paired-binary representation (2B), which would result in an increase of $C$ throughout successive layers.
> > > > > > > >
> > > > > > > > In the comparison with Reing our argument is that the measures by Reing et al., although intended to measure a similar quantity, do not observe this decrease through successive layers that our approach picks up on. As a minor remark, please note that the ordering of the layers for $C_\text{Reing}$ is violated not in the first epochs, but rather throughout most of the remaining epochs.

---

> > > > ### Comment · Reviewer_MWVV · 2023-03-02
> > > > **Predictability and operational meaning**
> > > >
> > > > The main power of information theory lies in the fact that operational problems can be solved
> > > > (sometimes completely characterised) via theoretical arguments. The most successful
> > > > (in terms of applicability) information measures are those that have been defined $\textit{operationally}$.
> > > > I.e., given an operational problem (e.g., compression) defined $\textbf{rigorously}$, in mathematical terms,
> > > > if the information measure (e.g., Shannon's entropy) appears as a fundamental limit of the problem
> > > > (e.g., the expected length of the UD encoding is always larger than entropy of the source AND is at most entropy + 1)
> > > > $\textbf{then}$ the information measure has operational significance. Renyi's entropy is known
> > > > to be connected to similar encodings (uniquely decodable) but the 'cost' of the encoding
> > > > is measured differently (see 'A coding theorem and Rényi's entropy'
> > > > by Campbell), or has been linked to (a specific and rigorously defined notion of) guessing by
> > > > Arikan (An inequality on guessing and its application to sequential decoding, Arikan 1996).
> > > > If you say "predictability" without defining what predicting is, then I cannot agree on a link between
> > > > MI and predictability. What you described, while intuitive, is still $\textbf{not}$ rigorous. First, we need
> > > > to agree on a notion of predictability then, we need to mathematically relate the notion of predictability
> > > > to MI and only then I could agree. What you described could easily be explained as well by considering
> > > > the total variation between point-wise conditionals and marginals. Instead of ratios, we look at differences.
> > > > Or Renyi's Divergences with norms of ratios. $\textbf{Every}$ information measure is defined starting
> > > > from those very same ratios, some have a log outside, some have a square, and others have a max.
> > > > If the discussion you brought forth was a sound justification, they would all be equally linked to 'predictability'.
> > > >  Which information measure is truly linked to the problem? It depends on how we define it and what type
> > > > of bound one can prove. Fano's inequality is definitely a link between (a converse on)
> > > > estimation and Shannon measures, but not enough to argue 'they're the same thing'.

---

> > > > > ### Author Response · Authors · 2023-03-08
> > > > > **Predictability and operational meaning**
> > > > >
> > > > > We agree that predictability needs to be well defined in order to rigorously relate it to Mutual Information, however adding a section about this relation to the manuscript is only necessary if it will provide additional value to the reader. As we do not refer to predictability in the manuscript any longer, we suggest it might be enough to agree that Mutual Information is a very interesting and relevant notion of information that can be studied in DNNs. However, just to clarify our previous response: The ultimate goal of a classifier neural network is to correctly predict the label for a given input sample. Mathematically, this predictive ability of the network can be quantified by the *accuracy*, i.e. the percentage of correctly classified data samples, computed on the train or test set. Via Fano's inequality and the data processing inequality, one can thus identify a high Shannon Mutual Information between the target label and the hidden layer representation as a necessary criterion for good accuracy. We do not claim that Shannon Mutual Information is the only measure that can in principle be related in a similar way.

---

> > > ### Comment · Reviewer_MWVV · 2023-03-02
> > > **Information**
> > >
> > > " Furthermore, the widespread use of a cross-entropy loss function in DNNs, which itself is a concept related to Shannon information theory, further motivates that analyzing Shannon information in earlier layers might reveal interesting facts about the learning dynamics"
> > >
> > > I never disagreed with the fact that studying Mutual Information in DNNs was interesting, I am disputing the way the word information and predictability were thrown around in a scientific paper, where definitions and language should be in most cases dry, rigorous, and precise.
> > >
> > > As said before, I do appreciate the effort, but I still disagree with many points in the manuscript.
> > > These are perhaps not crucial for this venue so I will leave them as hints for the authors to look into
> > > rather than explicit requests, especially since they are in the introductory part:
> > > -  I can guess what you mean with this sentence 'syntactical measures' as opposed to 'semantic'
> > >    but anyone working on information measures would find 'syntactical' a very unusual attribute;
> > > -  while the case considered here is such that MI = Shannon Entropy,  I don't see why one should replace,
> > >    in a more general setting, 'Mutual Information' with Renyi or Tsallis (which anyway are
> > >   essentially equivalent as one is a monotone mapping of the other) but rather with other information
> > >   measures like Arimoto's or Sibson's MI, or simply any f-Divergence between joint and product of
> > >   the marginals;
> > > - "Yet, even with such methodological problems out of the way, the question posed above on how features
> > >    are represented cannot be answered using classic information theory alone." - I continue to believe and state
> > >    that this is $\textbf{not}$ a limitation of Information Theory but rather (and $\textit{perhaps}$) a limitation of
> > >    Shannon's Mutual Information in particular (I still am not fully convinced that this decomposition is effectively
> > >    needed and would love to see a setting where PID does more than MI can).
> > >
> > > "This said, the idea of using information theory to quantify the information
> > > in the activations of Deep Neural Network (DNN) hidden layers has been met with fierce debate after
> > > methodological flaws surfaced (Saxe et al., 2019; Goldfeld et al., 2019) in the influential early works by Tishby
> > > et al. (Tishby and Zaslavsky, 2015; Shwartz-Ziv and Tishby, 2017). We address these issues in this work and
> > > show how information theory can be applied to quantized networks in a theoretically sound way."
> > >
> > > It might be worth arguing what these methodological flaws are and how the will be addressed in the work.

---

> > > > ### Author Response · Authors · 2023-03-08
> > > > **Information (1/2)**
> > > >
> > > > >I can guess what you mean with this sentence 'syntactical measures' as opposed to 'semantic' but anyone working on information measures would find 'syntactical' a very unusual attribute;
> > > >
> > > > While the term 'syntactical' information has been used before in the literature to refer to non-semantical information, we acknowledge that this terminology is not sufficiently widespread within the community and have removed it from our manuscript to avoid unnecessary confusion.
> > > >
> > > > >while the case considered here is such that MI = Shannon Entropy, I don't see why one should replace, in a more general setting, 'Mutual Information' with Renyi or Tsallis (which anyway are essentially equivalent as one is a monotone mapping of the other) but rather with other information measures like Arimoto's or Sibson's MI, or simply any $f$-Divergence between joint and product of the marginals;
> > > >
> > > > To start with, mutual information posesses several unique properties which other $f$-Divergences do not have, such as additivity of MI for independent variables (Mutual information is the unique solution when taking these properties into consideration derived by Robert Fano in *Transmission of information, p.31*). While it is not immediately obvious why these are necessary desiderata for the analysis of DNNs, these intuitive properties further support the choice of Shannon Information over arbitrary other $f$-Divergences in the absence of strong arguments of why to prefer another.
> > > >
> > > > On a more practical note, the partial information decomposition framework is already established for shannon mutual informaiton and provides a systematic way to look beyond pairwise relations, that shannon mutual information provide. The structure of partial information decomposition is indeed relatively agnostic to the choice of a specific information measure, since it is built on mereological (parthood) arguments and therefore can be derived for other information measures such as $f$-Divergences. However, one needs to define a consistent notion of *redundancy* for these quantities and make sure these redundancies adhere to the properties outlined by Gutknecht et al. To our knowledge, so far the only application of the PID framework to a quantity other than mutual information can be found in Partial *Entropy* Decomposition - still within the framework of Shannon Information - while the application of a PID-like decomposition to other notions of information remains an interesting unexplored area of research to this date.
> > > >
> > > > Nevertheless, as a future line of research it might be interesting to investigate whether using other $f$-Divergences as the basis of a PID yields not only quantitatively but also qualitatively different results.

---

> > > > ### Author Response · Authors · 2023-03-08
> > > > **Information (2/2)**
> > > >
> > > > >"Yet, even with such methodological problems out of the way, the question posed above on how features are represented cannot be answered using classic information theory alone." - I continue to believe and state that this is not a limitation of Information Theory but rather (and perhaps) a limitation of Shannon's Mutual Information in particular
> > > > >(I still am not fully convinced that this decomposition is effectively needed and would love to see a setting where PID does more than MI can).
> > > >
> > > > We believe that the underlying problem of not being able to differentiate between synergistic and redundant contributions is not unique to Shannon Mutual Information but rather a shortcoming that all *pairwise* information measures (i.e. measures that quantify some sort of correlation between two variables, which themselves might be composites of multiple variables) have in common. This can be seen from the fact that the arguments put forth in Section 3.1 about the underdeterminedness of the PID problem do not rely on any property particular to Shannon Mutual Information but hold true more generally. In fact, the whole structure of the PID redundancy lattice (Appendix A.1.3) rests only on the mereology axiom that the "information" (however one defines this term) that a target variable has with a set of source variables is always contained in the "information" that that target variable has with a superset of the previous source set (See Gutknecht et al.).
> > > > On the other hand, if one has a *multivariate* information measure quantifying some notion of synergy or redundancy, we believe that because of the universality of the arguments with which the PID lattice structure is established, one should expect this quantity to be consistent with this lattice. Since there might very well be shortcuts to compute these quantities not involving PID atoms, however, (as it would be the case for Representational Complexity with a Synergy-based measure), this connection to the PID lattice might not always be immediately apparent.
> > > > It doesn't come up as a surprise that the many early attempts to quantify synergy and redundacy where exposed by PID to be conflating synergy and redundancy such as the interaction information and any of its derivatives. Similarly, transfer entropy can be better understood with PID as consiting of two parts: the state dependent and state independent transfer entropy (*Williams, P.L.; Beer, R.D. Generalized Measures of Information Transfer*) Also expanding PID to $\Phi$ID by Mediano et al. shows the generality of PID in capturing part-whole relationships (as an example Table I in *Beyond integrated information: A taxonomy of information dynamics phenomena*)
> > > > Nevertheless, we agree that the statement in the paper could be made more precise by referring only to the pairwise measure of Shannon Mutual Information and have made changes accordingly.
> > > >
> > > >
> > > > >"This said, the idea of using information theory to quantify the information in the activations of Deep Neural Network (DNN) hidden layers has been met with fierce debate after methodological flaws surfaced (Saxe et al., 2019; Goldfeld et al., 2019) in the influential early works by Tishby et al. (Tishby and Zaslavsky, 2015; Shwartz-Ziv and Tishby, 2017). We address these issues in this work and show how information theory can be applied to quantized networks in a theoretically sound way."
> > > > >It might be worth arguing what these methodological flaws are and how the will be addressed in the work.
> > > >
> > > > As explained in Section 2, the issue with the approach of Tishby et al. is that they claim to estimate Shannon Mutual Information quantities between continuous variables which have a deterministic relationship between them. Due to this continuous relationship, however, this mutual information is almost always infinite, and what they are measuring with their binning approach can at best be reinterpreted as measures of geometric clustering. In our approach, we use a strong quantization within the network to intrinsically limit the channel capacity of the network to make mutual information quantities well-defined and meaningful.
> > > >
> > > > We added a reference to Section 2 in the introduction to make it easier to follow our arguments. We opted not to expand the brief mention of the issue in the introduction as we feel it would negatively impact the flow of reading.

---

> > ### Comment · Reviewer_MWVV · 2023-03-02
> > **Harshness**
> >
> > Apologies if the review felt 'harsh', I was not trying to be. Being precise and dry has its disadvantages.
> > I appreciate the removal of "rigorous" and the additional statements introduced to justify the term
> > "information" as well as the general effort put to improve the quality of the paper and make it more
> > accessible.

---

> > > ### Author Response · Authors · 2023-03-08
> > > **General Feedback**
> > >
> > > We thank the reviewer for the additional clarifications and points raised - we highly value the effort to help improve our manuscript. In subsequent comments, we respond to each of the points and incorporated changes into the manuscript accordingly.

---

### Review · Reviewer_jPSX · 2023-02-08

**Summary Of Contributions:**

This paper uses partial information decomposition in order to define and then measure a measure of representational complexity.  Intuitively this measure is meant to capture how many neurons need to be observed in order to extract useful information.  For something like a one hot encoding you'd expect their measure of representational complexity to be 1, and would go up as a representation becomes more distributed.  This measure itself is defined by a weighted sum over all of the terms in a partial information decomposition, something that grows super exponentially.

The paper then measures their proposed complexity measure on some simple networks and shows that the representational complexity decays over time, this seems to occur in a few different networks.  Given computational challenges, their measure can't be computed exactly on anything more than 5 neurons, but for larger networks they discuss and present some approximations which back up the other findings.

**Audience:**

Yes

**Broader Impact Concerns:**

None.

**Claims And Evidence:**

Yes

**Requested Changes:**

Please include version of the figures that report the representational complexity on the training set as well as on the test set, I'm curious whether there are any differences there.

Please also seriously consider including a scenario in which the representational complexity *doesn't* decrease over time, either by inserting some kind of strange fixed bijection at the readout layer or by some other means.  I think that would add a lot to start to formulate hypothesis about why this behavior might be happening or what implications it might have.

**Strengths And Weaknesses:**

Overall, I think the paper is well written, and has an audience at TMLR.

I felt as though the paper did a good job motivating the measure of representational capacity and giving enough background that people could follow and have a good sense of what was being measured.

The paper then reports on the measurements they took, for, admittedly small networks, but they acknowledge this and also use approximation schemes to measure larger networks as well.  All of the empirical results suggest and validate the claim that their measure of representational capacity is decreasing over time.

In terms of weaknesses.  I suppose one thing I would have liked to have seen is whether it would be possible to break this observation.  For instance, if we expect that the final prediction is classification for which we expect a one-hot like set of activations at the logit layer, I would perhaps naturally assume the earlier layers would also be driven towards lower representational capacity.  It seems like one way you could try to break this would be to insert a fixed random rotation matrix between the readout logits and the resulting class prediction.  I.e. force the final predictions to be distributed.  (I'm assuming here that such a representation would necessarily have a higher representational capacity, but am not certain about that).

As it stands the paper introduces a new measure and measures it and sees that it shows the same behavior across several different networks, which I can't really complain about, but also feel a bit as though I'm left not understanding what the implications or inferences I should draw from those observations are.  I don't really have any sense of why this should happen, or under which conditions I shouldn't expect it to happen.  If you could add some experiment, toy or otherwise that demonstrated different dynamics for the representational complexity, it seems like that would add a lot to the paper.

---

> ### Author Response · Authors · 2023-02-22
> **Addressing weaknesses and supplying additional empirical findings**
>
> We thank the reviewer for his efforts in understanding our approach and especially for proposing very interesting subsequent ideas. Following this feedback, we added multiple additional experiments and tried to increase the basis for forming hypotheses about the dynamics of the representational complexity in various scenarios. In the following, we will respond to the weaknesses and requested changes.
>
> > In terms of weaknesses. I suppose one thing I would have liked to have seen is whether it would be possible to break this observation. For instance, if we expect that the final prediction is classification for which we expect a one-hot like set of activations at the logit layer, I would perhaps naturally assume the earlier layers would also be driven towards lower representational capacity. It seems like one way you could try to break this would be to insert a fixed random rotation matrix between the readout logits and the resulting class prediction. I.e. force the final predictions to be distributed. (I'm assuming here that such a representation would necessarily have a higher representational capacity, but am not certain about that).
>
> We increased our efforts in trying to "break" the trend of decreasing representational complexity over the course of training. Additional to the network where we enforced a binary output representation (for which one might expect a higher representational complexity throughout the network, not only in the last layer), we tested a setup with a rotation matrix between the last layer and the softmax output. This change, however, did also not change the trend of decreasing representational complexity over training in a significant way, possibly because this rotation can just be undone by the last linear layer; potentially also because even in a rotated scenario, individual neurons still reveal a lot of information about the label. Instead of the rotation example, we included an experiment where the network was trained on shuffled labels, forcing the network to overfit on individual samples of the train set (see further below for more explanation).
>
> > As it stands the paper introduces a new measure and measures it and sees that it shows the same behavior across several different networks, which I can't really complain about, but also feel a bit as though I'm left not understanding what the implications or inferences I should draw from those observations are. I don't really have any sense of why this should happen, or under which conditions I shouldn't expect it to happen. If you could add some experiment, toy or otherwise that demonstrated different dynamics for the representational complexity, it seems like that would add a lot to the paper.
>
> >Requested Changes:
> Please include version of the figures that report the representational complexity on the training set as well as on the test set, I'm curious whether there are any differences there.
>
> > Please also seriously consider including a scenario in which the representational complexity doesn't decrease over time, either by inserting some kind of strange fixed bijection at the readout layer or by some other means. I think that would add a lot to start to formulate hypothesis about why this behavior might be happening or what implications it might have.
>
> We followed the requested change of the reviewer and added separate results computed for the train and test datasets, and report a trend of higher complexity on the test set towards the end of training.
>
> For the experiment with shuffled labels mentioned above, we observe a delayed increase in accuracy coinciding with a temporary peak in the representational complexity for both the datasets. After this peak however, the representational complexity for the training set drops again, while it stayed substantially higher for the test set. We believe that this elevated complexity on the test set is related to the network overfitting on train samples, leading to a difference in the representation for samples that the network has trained on compared to unseen samples. However, we are hesitant to draw any unfounded conclusions and point to further research for the investigation of this phenomenon.
>
> We hope that, overall, these additions and the comparison with other comparable approaches help the reader to gain some intuition about the behaviour of representational complexity under different circumstances and help to put our work more into context.

---

### Decision · Action_Editors · 2023-04-10

**Recommendation:** Accept as is

**Comment:**

This paper defines a novel rich measure of complexity quantifying the representational power of subsets of a neural net nodes towards a target (label) variable. It also provides small scale experiments computing it across layers for very small networks. Approximation schemes are proposed for slightly larger experiments, together with additional numerics. Some comparisons with similar notions found in the literature are discussed.
The paper has generated a great deal of meaningful discussions with the reviewers, which I'd like to thank. In particular, the lack of "rigorous and precise statements" or the fuzzyness of certain notions has been emphasized, leading to improvements. I got the impression that numerous points raised by the reviewers were adressed, at least at some level.
From my own reading, I got on the one hand the same impression as all reviewers, of a paper containing interesting original concepts. Also reviewer mLwm, despite suggesting a reject, pointed "The mathematical definition of the proposed representational complexity makes a lot of sense; I personally think this quantity has some potential to be a useful tool for the interpretation of the neural network architecture." and I agree with the statement. On the other hand, it is also true that after reading some concepts / notions remain hard to really grasp and quite elusive (it took me some time to even understand the pre-requisite of what an atom is). Yet, I ended with the impression that the paper will spark the interest of some readers and may lead to interesting follow up. The writing could be a bit more rigorous to ease the reading, I believe, yet I think this style may fit others. I think that it is important for the authors to publish the final link to the codes to make the notions more "concrete" and, importantly, reproducible, especially for other specialists in the field to compare to their own favorite complexity notion (as the comparison work done here remains partial). A curiosity: I also wonder whether links could be established with refs https://www.mdpi.com/1099-4300/20/10/739, https://arxiv.org/pdf/2008.00520.pdf . Given all this, I believe that the paper is of sufficient interest to be published as such.

**Audience:**

This paper has good chances to pick the interest of the TMLR audience, yes, in particular those looking for new conceptual approaches (rather than truly practical, at the moment) for the quantification of information flows in neural nets. I was wondering whether this paper would better fit an IEEE journal, but I think that both choices make sense for that paper. The style of the present contribution, which is more informal than IEEE standards, is also I guess a deliberate choice of the authors in targeting a certain audience.

**Claims And Evidence:**

As pointed by the reviewers, evidence are scarce. But this is also related to the intrinsic combinatorial difficulty of computing the proposed representational complexity measure. Some small scale experiments are provided, which are going in the direction of the intuitive explanations provided by the authors. I tend to agree with reviewer MWVV that this contribution does not provide very strong numerical evidence, nor strong theory. Yet, I believe that the evidence are sufficient to trigger further research in the challenging and active research direction concerned with the quantification of the representation learning in neural nets.